# ON THE "STEERABILITY" OF GENERATIVE ADVERSARIAL NETWORKS

**Ali Jahanian\*, Lucy Chai\*, & Phillip Isola**
Massachusetts Institute of Technology
Cambridge, MA 02139, USA
{`jahanian,lrchai,phillipi`}@mit.edu

## ABSTRACT

An open secret in contemporary machine learning is that many models work beautifully on standard benchmarks but fail to generalize outside the lab. This has been attributed to biased training data, which provide poor coverage over real world events. Generative models are no exception, but recent advances in generative adversarial networks (GANs) suggest otherwise – these models can now synthesize strikingly realistic and diverse images. Is generative modeling of photos a solved problem? We show that although current GANs can fit standard datasets very well, they still fall short of being comprehensive models of the visual manifold. In particular, we study their ability to fit simple transformations such as camera movements and color changes. We find that the models reflect the biases of the datasets on which they are trained (e.g., centered objects), but that they also exhibit some capacity for generalization: by "steering" in latent space, we can shift the distribution while still creating realistic images. We hypothesize that the degree of distributional shift is related to the breadth of the training data distribution. Thus, we conduct experiments to quantify the limits of GAN transformations and introduce techniques to mitigate the problem. Code is released on our project page: `https://ali-design.github.io/gan_steerability/`.

## 1 INTRODUCTION

The quality of deep generative models has increased dramatically over the past few years. When introduced in 2014, Generative Adversarial Networks (GANs) could only synthesize MNIST digits and low-resolution grayscale faces (Goodfellow et al., 2014). The most recent models, however, produce diverse high-resolution images that are often indistinguishable from natural photos (Brock et al., 2018; Karras et al., 2018).

Science fiction has long dreamed of virtual realities filled of synthetic content as rich as, or richer, than the real world (e.g., *The Matrix*, *Ready Player One*). How close are we to this dream? Traditional computer graphics can render photorealistic 3D scenes, but cannot automatically generate detailed content. Generative models like GANs, in contrast, can create content from scratch, but we do not currently have tools for navigating the generated scenes in the same kind of way as you can walk through and interact with a 3D game engine.

In this paper, we explore the degree to which you can navigate the visual world of a GAN. Figure 1 illustrates the kinds of transformations we explore. Consider the dog at the top-left. By moving in some direction of GAN latent space, can we hallucinate walking toward this dog? As the figure indicates, and as we will show in this paper, the answer is yes. However, as we continue to zoom in, we quickly reach limits. Once the dog face fills the full frame, continuing to walk in this direction fails to increase the zoom. A similar effect occurs in the daisy example (row 2 of Fig. 1), where a direction in latent space moves the daisy up and down, but cannot move it out of frame.

We hypothesize that these limits are due to biases in the distribution of images on which the GAN is trained. For example, if the training dataset consists of centered dogs and daises, the same may be the case in GAN-generated images. Nonetheless, we find that *some* degree of transformation is possible. When and why can we achieve certain transformations but not others?

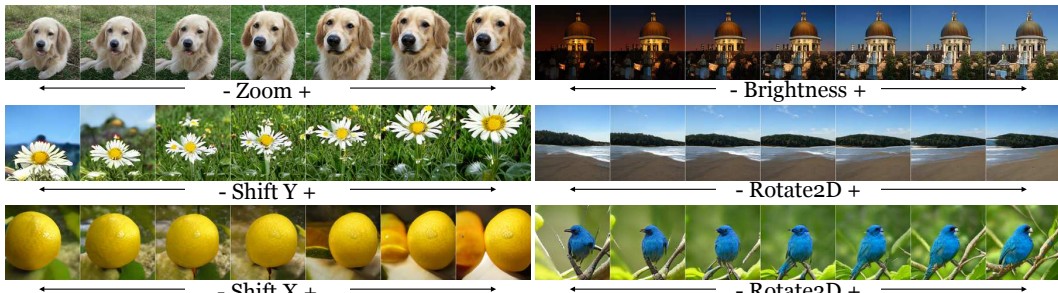

Figure 1: Learned latent space trajectories in generative adversarial networks correspond to visual transformations like camera shift and zoom. Take the "steering wheel", drive in the latent space, and explore the natural image manifold via generative transformations!

This paper seeks to quantify the degree to which we can achieve basic visual transformations by navigating in GAN latent space. In other words, are GANs "steerable" in latent space?[1] We analyze the relationship between the data distribution on which the model is trained and the success in achieving these transformations. From our experiments, it is possible to shift the distribution of generated images to some degree, but we cannot extrapolate entirely out of the dataset's support. In particular, attributes can be shifted in proportion to the variability of that attribute in the training data. We further demonstrate an approach to increase model steerability by jointly optimizing the generator and latent direction, together with data augmentation on training images. One of the current criticisms of generative models is that they simply interpolate between datapoints, and fail to generate anything truly new, but our results add nuance to this story. It *is* possible to achieve distributional shift, but the ability to create realistic images from a modified distributions relies on sufficient diversity in the dataset along the dimension that we vary.

Our main findings are:

- A simple walk in the latent space of GANs achieves camera motion and color transformations in the output image space. These walks are learned in self-supervised manner without labeled attributes or distinct source and target images.

- The linear walk is as effective as more complex non-linear walks, suggesting that the models learn to roughly linearize these operations without being explicitly trained to do so.

- The extent of each transformation is limited, and we quantify a relationship between dataset variability and how much we can shift the model distribution.

- The transformations are a general-purpose framework that work with different model architectures, e.g. BigGAN, StyleGAN, and DCGAN, and illustrate different disentanglement properties in their respective latent spaces.

- Data augmentation improves steerability, as does jointly training the walk trajectory and the generator weights, which allows us to achieve larger transformation effects.

## 2  RELATED WORK

Latent space manipulations can be seen from several perspectives – how we achieve it, what limits it, and what it enables us to do. Our work addresses these three aspects together, and we briefly refer to each one in related work.

**Interpolations in latent space** Traditional approaches to image editing with GAN latent spaces find linear directions that correspond to changes in labeled attributes, such as smile-vectors and gender-vectors for faces (Radford et al., 2015; Karras et al., 2018). However these manipulations are not exclusive to GANs; in flow-based generative models, linearly interpolating between two encoded images allow one to edit a source image toward attributes of the target (Kingma & Dhariwal, 2018). Möllenhoff & Cremers (2019) proposes a modified GAN formulation by treating data

---

[1]We use the term "steerable" in analogy to the classic steerable filters of Freeman & Adelson (1991).

as directional $k$-currents, where moving along tangent planes naturally corresponds to interpretable manipulations. Upchurch et al. (2017) removes the generative model entirely and instead interpolates in the intermediate feature space of a pretrained classifier, again using feature mappings of source and target sets to determine an edit direction. Unlike these approaches, we learn our latent-space trajectories in a self-supervised manner without labeled attributes or distinct source and target images. Instead, we learn to approximate editing operations on individual source images. We find that linear trajectories in latent space can capture simple image manipulations, e.g., zoom-vectors and shift-vectors, although we also obtain similar results using nonlinear trajectories.

**Dataset bias** Biases from training data and network architecture both impact the generalization capacity of learned models (Torralba & Efros, 2011; Geirhos et al., 2018; Amini et al.). Dataset biases partly comes from human preferences in taking photos: we tend to take pictures in specific "canonical" views that are not fully representative of the entire visual world (Mezuman & Weiss, 2012; Jahanian et al., 2015). Consequently, models trained with these datasets inherit their biases. This may result in models that misrepresent the given task – such as tendencies towards texture bias rather than shape bias on ImageNet classifiers (Geirhos et al., 2018) – and in turn limits their generalization performance on similar objectives (Azulay & Weiss, 2018). Our latent space trajectories transform the output corresponding to various image editing operations, but ultimately we are constrained by biases in the data and cannot extrapolate arbitrarily far beyond the data's support.

**Generative models for content creation** The recent progress in generative models has opened interesting avenues for content creation (Brock et al., 2018; Karras et al., 2018), including applications that enable users to fine-tune the generated output (Simon; Zhu et al., 2016; Bau et al., 2018). A by-product the current work is enable users to modify image properties by turning a single knob – the magnitude of the learned transformation in latent space. We further demonstrate that these image manipulations are not just a simple creativity tool; they also provide us with a window into biases and generalization capacity of these models.

**Applications of latent space editing** Image manipulations using generative models suggest several interesting downstream applications. For example, Denton et al. (2019) learns linear walks corresponding to various facial characteristics – they use these to measure biases in facial attribute detectors, whereas we study biases in the generative model that originate from training data. Shen et al. (2019) also assumes linear latent space trajectories and learns paths for face attribute editing according to semantic concepts such as age and expression, thus demonstrating disentanglement of the latent space. White (2016) suggests approaches to improve the learned manipulations, such as using spherical linear interpolations, resampling images to remove biases in attribute vectors, and using data augmentation as a synthetic attribute for variational autoencoders. Goetschalckx et al. (2019) applies a linear walk to achieve transformations corresponding to cognitive properties of an image such as memorability, aesthetics, and emotional valence. Unlike these works, we do not require an attribute detector or assessor function to learn the latent space trajectory, and therefore our loss function is based on image similarity between source and target images. In addition to linear walks, we explore using non-linear walks parametrized by neural networks for editing operations.

## 3 METHOD

Generative models such as GANs (Goodfellow et al., 2014) learn a mapping function $G$ such that $G : z \rightarrow x$. Here, $z$ is the latent code drawn from a Gaussian density and $x$ is an output, e.g., an image. Our goal is to achieve transformations in the output space by moving in latent space, as shown in Fig. 2. In general, this goal also captures the idea in equivariance, in which transformations in the input space result in equivalent transformations in the output space (c.f. Hinton et al. (2011); Cohen et al. (2019); Lenc & Vedaldi (2015)).

**Objective** We want to learn an $N$-dimensional vector representing the optimal path in latent space for a given transformation. The vector is multiplied with continuous parameter $\alpha$ which signifies the step size: large $\alpha$ values correspond to a greater degree of transformation, while small $\alpha$ values correspond to a lesser degree. Formally, we learn the walk $w$ by minimizing the objective function:

$$w^* = \arg\min_w \mathbb{E}_{z,\alpha}[\mathcal{L}(G(z+\alpha w), \texttt{edit}(G(z), \alpha))]. \tag{1}$$

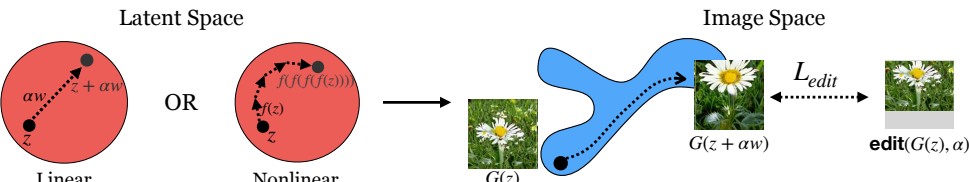

Figure 2: We aim to find a path in $z$ space to transform the generated image $G(z)$ to its edited version $\texttt{edit}(G(z, \alpha))$, e.g., an $\alpha\times$ zoom. This walk results in the generated image $G(z + \alpha w)$ when we choose a linear walk, or $G(f(f(...(z))))$ when we choose a non-linear walk.

Here, $\mathcal{L}$ measures the distance between the generated image after taking an $\alpha$-step in the latent direction $G(z + \alpha w)$ and the target $\texttt{edit}(G(z), \alpha)$ derived from the source image $G(z)$. We use $L2$ loss as our objective $\mathcal{L}$, however we also obtain similar results when using the LPIPS perceptual image similarity metric (Zhang et al., 2018) (see Appendix B.4.1). Note that we can learn this walk in a fully self-supervised manner – we perform the $\texttt{edit}(\cdot)$ operation on an arbitrary generated image and subsequently the vector to minimize the objective. Let $\texttt{model}(\alpha)$ denote the optimized transformation vector $w^*$ with the step size $\alpha$, defined as $\texttt{model}(\alpha) = G(z + \alpha w^*)$.

The previous setup assumes linear latent space walks, but we can also learn non-linear trajectories in which the walk direction depends on the current latent space position. For the non-linear walk, we learn a function, $f^*(z)$, which corresponds to a small $\epsilon$-step transformation $\texttt{edit}(G(z), \epsilon)$. To achieve bigger transformations, we apply $f$ recursively, mimicking discrete Euler ODE approximations. Formally, for a fixed $\epsilon$, we minimize

$$\mathcal{L} = \mathbb{E}_{z,n}[||G(f^n(z)) - \texttt{edit}(G(z), n\epsilon))||], \tag{2}$$

where $f^n(\cdot)$ is an $n$th-order function composition $f(f(f(...)))$, and $f(z)$ is parametrized with a neural network. We discuss further implementation details in Appendix A.4. We use this function composition approach rather than the simpler setup of $G(z + \alpha\text{NN}(z))$ because the latter learns to ignore the input $z$ when $\alpha$ takes on continuous values, and is thus equivalent to the previous linear trajectory (see Appendix A.3 for further details).

**Quantifying Steerability**   We further seek to quantify how well we can achieve desired image manipulations under each transformation. To this end, we compare the distribution of a given attribute, e.g., "luminance", in the dataset versus in images generated after walking in latent space.

For color transformations, we consider the effect of increasing or decreasing the $\alpha$ coefficient corresponding to each color channel. To estimate the color distribution of model-generated images, we randomly sample $N = 100$ pixels per image both before and after taking a step in latent space. Then, we compute the pixel value for each channel, or the mean RGB value for luminance, and normalize the range between 0 and 1.

For zoom and shift transformations, we rely on an object detector which captures the central object in the image class. We use a *MobileNet-SSD v1* (Liu et al., 2016) detector to estimate object bounding boxes, and average over image classes recognizable by the detector. For each successful detection, we take the highest probability bounding box corresponding to the desired class and use that to quantify the amount of transformation. For the zoom operation, we use the area of the bounding box normalized by the area of the total image. For shift in the X and Y directions, we take the center X and Y coordinates of the bounding box, and normalize by image width or height.

Truncation parameters in GANs (as used in Brock et al. (2018); Karras et al. (2018)) trade off between the diversity of the generated images and sample quality. When comparing generated images to the dataset distribution, we use the largest possible truncation for the model and perform similar cropping and resizing of the dataset as done during model training (see Brock et al. (2018)). When comparing the attributes of generated distributions under different $\alpha$ magnitudes to each other but not to the dataset, we reduce truncation to 0.5 to ensure better performance of the object detector.

**Reducing Transformation Limits**   Equations 1 and 2 learn a latent space walk assuming a pretrained generative model, thus keeping the model weights fixed. The previous approach allows us

to understand the latent space organization and limitations in the model's transformation capacity. To overcome these limits, we explore adding data augmentation by editing the training images with each corresponding transformation, and train the generative model with this augmented dataset. We also introduce a modified objective function that jointly optimizes the generator weights and a linear walk vector:

$$G^*, w^* = \arg\min_{G,w} \left( \mathcal{L}_{edit} + \mathcal{L}_{GAN} \right), \tag{3}$$

where the edit loss encourages low $L2$ error between learned transformation and target image:

$$\mathcal{L}_{edit} = L2 \left( G(z + \alpha w) - \texttt{edit}(G(z), \alpha) \right). \tag{4}$$

The GAN loss optimizes for discriminator error:

$$\mathcal{L}_{GAN} = \max_D \left( \mathbb{E}_{z,\alpha}[D(G(z + \alpha w))] - \mathbb{E}_{x,\alpha}[D(\texttt{edit}(x, \alpha))] \right), \tag{5}$$

where we draw images $x$ from the training dataset and perform data augmentation by applying the `edit` operation on them. This optimization approach encourages the generator to organize its latent space so that the transformations lie along linear paths, and when combined with data augmentation, results in larger transformation ranges which we demonstrate in Sec. 4.4

## 4 EXPERIMENTS

We demonstrate our approach using BigGAN (Brock et al., 2018), a class-conditional GAN trained on 1000 ImageNet categories. We learn a shared latent space walk by averaging across the image categories, and further quantify how this walk affects each class differently. We focus on linear walks in latent space for the main text, and show additional results on nonlinear walks in Sec. 4.3 and Appendix B.4.2. We also conduct experiments on StyleGAN (Karras et al., 2018), which uses an unconditional style-based generator architecture in Sec. 4.3 and Appendix B.5.

### 4.1 WHAT IMAGE TRANSFORMATIONS CAN WE ACHIEVE IN LATENT SPACE?

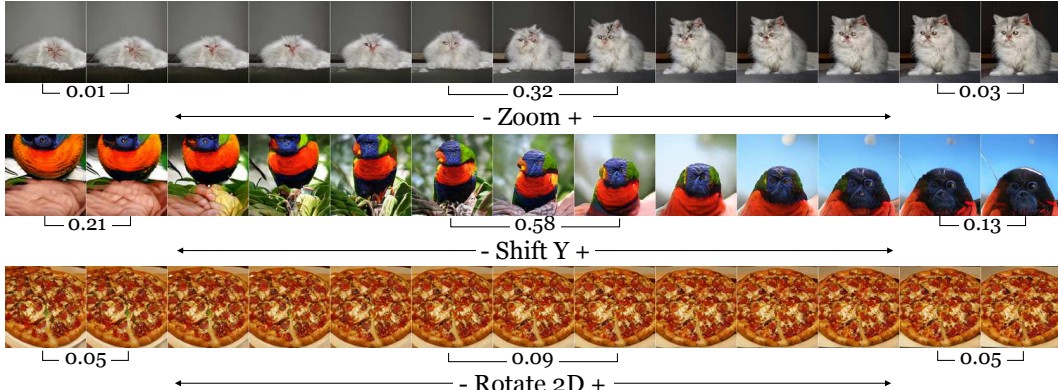

Figure 3: Transformation limits. As we increase the magnitude of $w^*$, the operation either does not transform the image any further, or the image becomes unrealisitic. Below each figure we also indicate the average LPIPS perceptual distance between 200 sampled image pairs of that category. Perceptual distance decreases as we move farther from the source (center image), which indicates that the images are converging.

We show qualitative results of the learned transformations in Fig. 1. By steering in the generator latent space, we learn a variety of transformations on a given source image (shown in the center panel of each transformation). Interestingly, several priors come into play when learning these image transformations. When we shift a daisy downwards in the Y direction, the model hallucinates that the sky exists on the top of the image. However, when we shift the daisy up, the model inpaints the remainder of the image with grass. When we alter the brightness of a image, the model transitions between nighttime and daytime. This suggests that the model can extrapolate from the original source image, and still remain consistent with the image context.

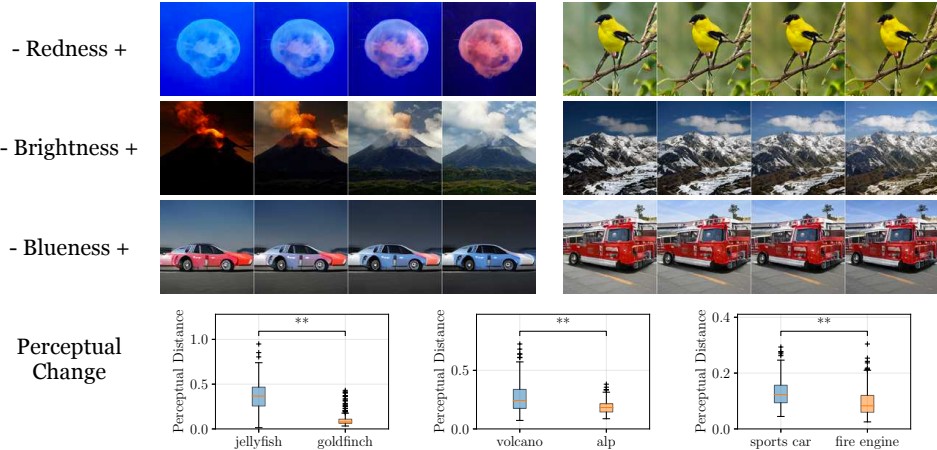

Figure 4: Each row shows how a single latent direction $w^*$ affects two different ImageNet classes. We observe that changes are consistent with semantic priors (e.g., "Volcanoes" explode, "Alps" do not). Boxplots show the LPIPS perceptual distance before and after transformation for 200 samples per class.

However, when we increase the step size of $\alpha$, we observe that the degree to which we can achieve each transformation is limited. In Fig. 3 we observe two potential failure cases: one in which the the image becomes unrealistic, and the other in which the image fails to transform any further. When we try to zoom in on a Persian cat, we observe that the cat no longer increases in size beyond some point, and in fact consistently undershoots the target zoom. On the other hand, when we try to zoom out on the cat, we observe that it begins to fall off the image manifold, and does not become any smaller after some point. Indeed, the perceptual distance (using LPIPS) between images decreases as we push $\alpha$ towards the transformation limits. Similar trends hold with other transformations: we are able to shift a lorikeet up and down to some degree until the transformation yields unrealistic output, and despite adjusting $\alpha$ on the rotation vector, we are unable to rotate a pizza. Are the limitations to these transformations governed by the training dataset? In other words, are our latent space walks limited because in ImageNet photos the cats are mostly centered and taken within a certain size? We seek to investigate and quantify these biases in the next sections.

An intriguing characteristic of the learned trajectory is that the amount it affects the output depends on the image class. In Fig. 4, we investigate the impact of the walk for different image categories under color transformations. By moving in the direction of a redness vector, we are able to successfully recolor a jellyfish, but we are unable to change the color of a goldfinch, which remains yellow which slight changes in background textures. Likewise, increasing brightness changes an erupting volcano to a dormant one, but does not have much effect on Alps, which only transitions between night and day. In the third example, we use our latent walk to turn red sports cars to blue, but it cannot recolor firetrucks. Again, perceptual distance over image samples confirms these qualitative observations: a 2-sample $t$-test yields $t = 20.77$, $p < 0.001$ for jellyfish/goldfinch, $t = 8.14$, $p < 0.001$ for volcano/alp, and $t = 6.84$, $p < 0.001$ for sports car/fire engine. We hypothesize that the different impact of the shared transformation on separate image classes relates to the variability in the underlying dataset. The overwhelming majority of firetrucks are red[2], but sports cars appear in a variety of colors. Therefore, our color transformation is constrained by the dataset biases of individual classes.

With shift, we can move the distribution of the center object by varying $\alpha$. In the underlying model, the center coordinate of the object is most concentrated at half of the image width and height, but after applying the shift in X and shift in Y transformation, the mode of the transformed distribution varies between 0.3 and 0.7 of the image width/height. To quantify the distribution changes, we compute the area of intersection between the original model distribution and the distribution after applying each transformation and observe that the intersection decreases as we increase or decrease the magnitude of $\alpha$. However, our transformations are limited to a certain extent – if we increase $\alpha$

---

[2]but apparently blue fire trucks do exist! (DiGrazia, 2019)

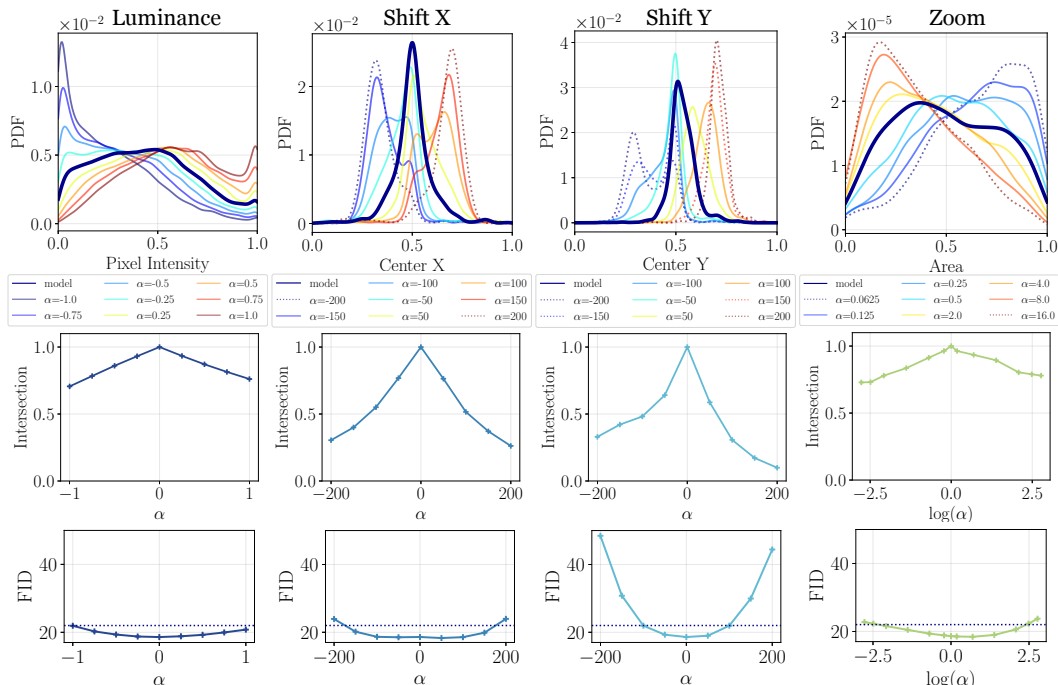

Figure 5: Quantifying the extent of transformations. We compare the attributes of generated images under the raw model output $G(z)$, compared to the distribution under a learned transformation model($\alpha$). We measure the intersection between $G(z)$ and model($\alpha$), and also compute the FID on the transformed image to limit our transformations to the natural image manifold.

beyond 150 pixels for vertical shifts, we start to generate unrealistic images, as evidenced by a sharp rise in FID and converging modes in the transformed distributions (Fig. 5 columns 2 & 3).

We perform a similar procedure for zoom, by measuring the area of the bounding box for the detected object under different magnitudes of $\alpha$. Like shift, we observe that subsequent increases in $\alpha$ magnitude start to have smaller and smaller effects on the mode of the resulting distribution (Fig. 5 last column). Past an 8x zoom in or out, we observe an increase in the FID signifying decreasing image quality. Interestingly for zoom, the FID under zooming in and zooming out is anti-symmetric, indicating that how well we can zoom-in and retain realisitic images differs from that of zooming out. These trends are consistent with the plateau in transformation behavior that we qualitatively observe in Fig. 3. Although we can arbitrarily increase the $\alpha$ step size, after some point we are unable to achieve further transformation and risk deviating from the natural image manifold.

## 4.2 How does the data affect the transformations?

Is the extent to which we can transform each class, as we observed in Fig. 4, due to limited variability in the underlying dataset for each class? One way of quantifying this is to measure the difference in transformed model means, model($+\alpha$) and model($-\alpha$), and compare it to the spread of the dataset distribution. For each class, we compute standard deviation of the dataset with respect to our statistic of interest (pixel RGB value for color, and bounding box area and center value for zoom and shift transformations respectively). We hypothesize that if the amount of transformation is biased depending on the image class, we will observe a correlation between the distance of the mean shifts and the standard deviation of the data distribution.

More concretely, we define the change in model means under a given transformation as:

$$\Delta\mu_k = \mu_{k,\text{model}(+\alpha^*)} - \mu_{k,\text{model}(-\alpha^*)} \tag{6}$$

for a given class $k$ and we set $\alpha^*$ to be largest and smallest $\alpha$ values used in training. The degree to which we achieve each transformation is a function of $\alpha$, so we use the same $\alpha$ value for all classes – one that is large enough to separate the means of $\mu_{k,\text{model}(\alpha^*)}$ and $\mu_{k,\text{model}(-\alpha^*)}$ under

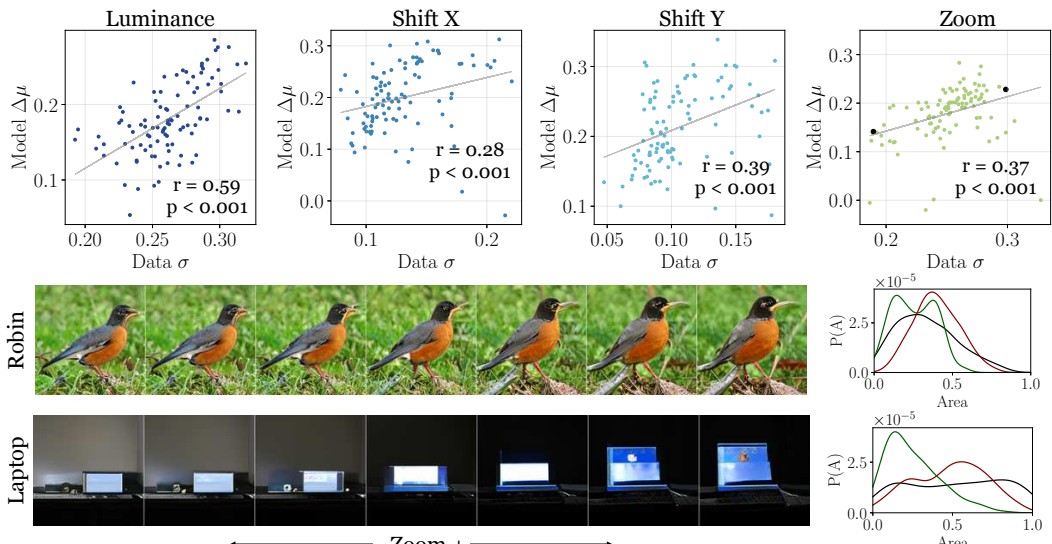

Figure 6: Understanding per-class biases. We observe a correlation between the variability in the training data for ImageNet classes, and our ability to shift the distribution under latent space transformations. Classes with low variability (e.g., robin) limit our ability to achieve desired transformations, in comparison to classes with a broad dataset distribution (e.g., laptop). To the right, we show the distribution of the zoom attribute in the dataset (black) and under $+\alpha$ (red) and $-\alpha$ (green) transformations for these two examples.

transformation, but also for which the FID of the generated distribution remains below a threshold $T$ of generating reasonably realistic images (for our experiments we use $T = 22$).

In Fig. 6 we plot the standard deviation $\sigma$ of the dataset on the x-axis, and the model $\Delta\mu$ under a $+\alpha^*$ and $-\alpha^*$ transformation on the y-axis, as defined in Eq. 6. We sample randomly from 100 classes for the color, zoom and shift transformations, and generate 200 samples of each class under the positive and negative transformations. We use the same setup of drawing samples from the model and dataset and computing the statistics for each transformation as described in Sec. 4.1.

Indeed, we find that the width of the dataset distribution, captured by the standard deviation of random samples drawn from the dataset for each class, relates to how much we can transform. There is a positive correlation between the spread of the dataset and the magnitude of $\Delta\mu$ observed in the transformed model distributions, and the slope of all observed trends differs significantly from zero ($p < 0.001$ for all transformations). For the zoom transformation, we show examples of two extremes along the trend. For the "robin" class the spread $\sigma$ in the dataset is low, and subsequently, the separation $\Delta\mu$ that we are able to achieve by applying $+\alpha^*$ and $-\alpha^*$ transformations is limited. On the other hand, for "laptops", the dataset spread is broad; ImageNet contains images of laptops of various sizes, and we are able to attain wider shifts in the model distribution.

From these results, we conclude that the amount of transformation we can achieve relates to the dataset variability. Consistent with our qualitative observations in Fig. 4, we find that if the images for a particular class have adequate coverage over the entire range of a given transformation, then we are better able to move the model distribution to both extremes. On the other hand, if the images for a given class are less diverse, the transformation is limited by this dataset bias.

## 4.3 ALTERNATIVE ARCHITECTURES AND WALKS

We ran an identical set of experiments using the nonlinear walk in the BigGAN latent space (Eq 2) and obtained similar quantitative results. To summarize, the Pearson's correlation coefficient between dataset $\sigma$ and model $\Delta\mu$ for linear walks and nonlinear walks is shown in Table 1, and full results in Appendix B.4.2. Qualitatively, we observe that while the linear trajectory undershoots the targeted level of transformation, it is able to preserve more realistic-looking results (Fig. 7). The

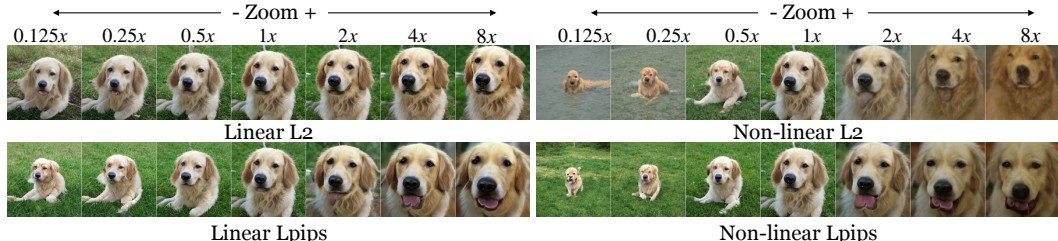

Figure 7: Comparison of linear and nonlinear walks for the zoom operation. The linear walk under-shoots the targeted level of transformation, but maintains more realistic output.

transformations involve a trade-off between minimizing the loss and maintaining realistic output, and we hypothesize that the linear walk functions as an implicit regularizer that corresponds well with the inherent organization of the latent space.

|  | Luminance | Shift X | Shift Y | Zoom |
|---|---|---|---|---|
| Linear | 0.59 | 0.28 | 0.39 | 0.37 |
| Non-linear | 0.49 | 0.49 | 0.55 | 0.60 |

Table 1: Pearson's correlation coefficient between dataset $\sigma$ and model $\Delta\mu$ for measured attributes. p-value for slope $< 0.001$ for all transformations.

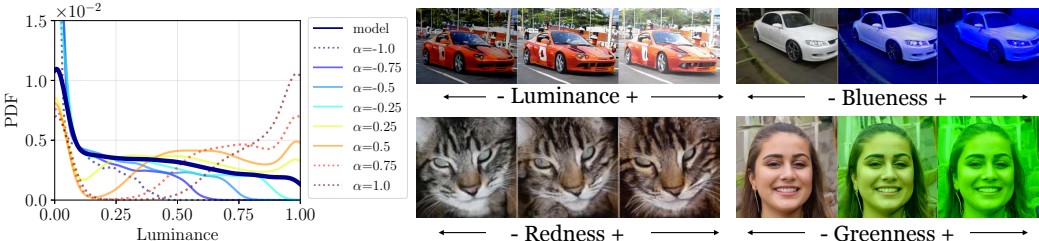

Figure 8: Distribution for luminance transformation learned from the StyleGAN cars generator, and qualitative examples of color transformations on various datasets using StyleGAN.

To test the generality of our findings across model architecture, we ran similar experiments on Style-GAN, in which the latent space is divided into two spaces, $z$ and $W$. As Karras et al. (2018) notes that the $W$ space is less entangled than $z$, we apply the linear walk to $W$ and show results in Fig. 8 and Appendix B.5. One interesting aspect of StyleGAN is that we can change color while leaving other structure in the image unchanged. In other words, while green faces do not naturally exist in the dataset, the StyleGAN model is still able to generate them. This differs from the behavior of BigGAN, where changing color results in different semantics in the image, e.g., turning a dormant volcano to an active one. StyleGAN, however, does not preserve the exact geometry of objects under other transformations, e.g., zoom and shift (see Appendix B.5).

## 4.4 TOWARDS STEERABLE GANS

So far, we have frozen the parameters of the generative model when learning a latent space walk for image editing, and observe that the transformations are limited by dataset bias. Here we investigate approaches to overcome these limitations and increase model steerability. For these experiments, we use a class-conditional DCGAN model (Radford et al., 2015) trained on MNIST digits (LeCun, 1998).

To study the effect of dataset biases, we train (1) a vanilla DCGAN and (2) a DCGAN with data augmentation, and then learn the optimal walk in Eq. 1 after the model has been trained – we refer to these two approaches in Fig. 9 as *argmin W* and *argmin W + aug*, respectively. We observe that adding data augmentation yields transformations that better approximate the target image and

attain lower $L2$ error than the vanilla DCGAN (blue and orange curves in Fig. 9). Qualitatively, we observe that transformations using the vanilla GAN (*argmin W*) become patchy and unrealistic as we increase the magnitude of $\alpha$, but when the model is trained with data augmentation (*argmin W + aug*), the digits retain their structural integrity.

Rather than learning the walk vector $w$ assuming a frozen generator, we may also jointly optimize the model and linear walk parameter together, as we formalized in Eq. 3. This allows the model to learn an equivariance between linear directions in the latent space and the corresponding image transformations. We refer to this model as *argmin G,W* in Fig. 9. Compared to the frozen generator (in *argmin W* and *argmin W + aug*), the joint objective further decreases $L2$ error (green curve in Fig. 9). We show additional qualitative examples in Appendix B.8. The steerable range of the generator increases with joint optimization and data augmentation, which provides additional evidence that training data bias impacts the models' steerability and generalization capacity. We tried DCGAN on CIFAR10 as a more complicated dataset, however were unable to get steering to be effective – all three methods failed to produce realistic transformations and joint training in fact performed the worst. Finding the right steering implementation per GAN and dataset, especially for joint training, may be a difficult problem and an interesting direction for future work.

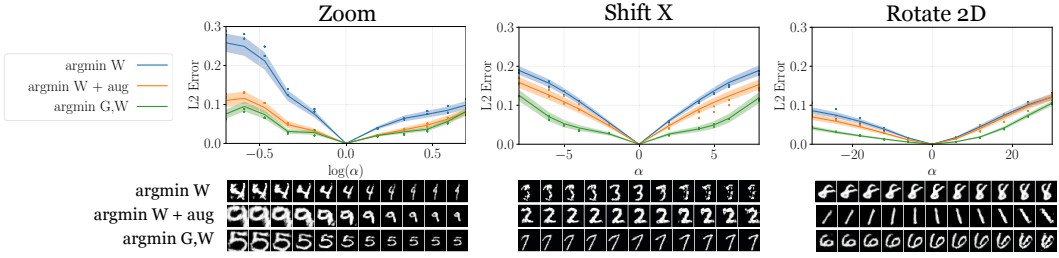

Figure 9: Reducing the effect of transformation limits. Using a DCGAN model on MNIST digits, we compare the $L2$ reconstruction errors on latent space walks for models trained with vanilla GANs without (*argmin W*) and with data augmentation (*argmin W + aug*). We also compare to jointly optimizing the generator and the walk parameters with data augmentation (*argmin G,W*), which achieves the lowest $L2$ error.

## 5 CONCLUSION

GANs are powerful generative models, but are they simply replicating the existing training data-points, or can they to generalize beyond the training distribution? We investigate this question by exploring walks in the latent space of GANs. We optimize trajectories in latent space to reflect simple image transformations in the generated output, learned in a self-supervised manner. We find that the model is able to exhibit characteristics of extrapolation – we are able to "steer" the generated output to simulate camera zoom, horizontal and vertical movement, camera rotations, and recolorization. However, our ability to naively move the distribution is finite: we can transform images to some degree but cannot extrapolate entirely outside the support of the training data. To increase model steerability, we add data augmentation during training and jointly optimize the model and walk trajectory. Our experiments illustrate the connection between training data bias and the resulting distribution of generated images, and suggest methods for extending the range of images that the models are able to create.

ACKNOWLEDGEMENTS

We would like to thank Quang H Le, Lore Goetschalckx, Alex Andonian, David Bau, and Jonas Wulff for helpful discussions. This work was supported by a Google Faculty Research Award to P.I., and a U.S. National Science Foundation Graduate Research Fellowship to L.C.

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

## A    METHOD DETAILS

### A.1    OPTIMIZATION FOR THE LINEAR WALK

We learn the walk vector using mini-batch stochastic gradient descent with the Adam optimizer (Kingma & Ba, 2014) in tensorflow, trained on 20000 unique samples from the latent space $z$. We share the vector $w$ across all ImageNet categories for the BigGAN model.

### A.2    IMPLEMENTATION DETAILS FOR LINEAR WALK

We experiment with a number of different transformations learned in the latent space, each corresponding to a different walk vector. Each of these transformations can be learned without any direct supervision, simply by applying our desired edit to the source image. Furthermore, the parameter $\alpha$ allows us to vary the extent of the transformation. We found that a slight modification to each transformation improved the degree to which we were able to steer the output space: we scale $\alpha$ differently for the learned transformation $G(z + \alpha_g w)$, and the target edit $\text{edit}(G(z), \alpha_t)$. We detail each transformation below:

**Shift.** We learn transformations corresponding to shifting an image in the horizontal X direction and the vertical Y direction. We train on source images that are shifted $-\alpha_t$ pixels to the left and $\alpha_t$ pixels to the right, where we set $\alpha_t$ to be between zero and one-half of the source image width or height $D$. When training the walk, we enforce that the $\alpha_g$ parameter ranges between -1 and 1; thus for a random shift by $t$ pixels, we use the value $\alpha_g = \alpha_t/D$. We apply a mask to the shifted image, so that we only apply the loss function on the visible portion of the source image. This forces the generator to extrapolate on the obscured region of the target image.

**Zoom.** We learn a walk which is optimized to zoom in and out up to four times the original image. For zooming in, we crop the central portion of the source image by some $\alpha_t$ amount, where $0.25 < \alpha_t < 1$ and resize it back to its original size. To zoom out, we downsample the image by $\alpha_t$ where $1 < \alpha_t < 4$. To allow for both a positive and negative walk direction, we set $\alpha_g = \log(\alpha_t)$. Similar to shift, a mask applied during training allows the generator to inpaint the background scene.

**Color.** We implement color as a continuous RGB slider, e.g., a 3-tuple $\alpha_t = (\alpha_R, \alpha_G, \alpha_B)$, where each $\alpha_R$, $\alpha_G$, $\alpha_B$ can take values between $[-0.5, 0.5]$ in training. To edit the source image, we simply add the corresponding $\alpha_t$ values to each of the image channels. Our latent space walk is parameterized as $z + \alpha_g w = z + \alpha_R w_R + \alpha_G w_G + \alpha_B w_B$ where we jointly learn the three walk directions $w_R$, $w_G$, and $w_B$.

**Rotate in 2D.** Rotation in 2D is trained in a similar manner as the shift operations, where we train with $-45 \leq \alpha_t \leq 45$ degree rotation. Using $R = 45$, scale $\alpha_g = \alpha_t / R$. We use a mask to enforce the loss only on visible regions of the target.

**Rotate in 3D.** We simulate a 3D rotation using a perspective transformation along the Z-axis, essentially treating the image as a rotating billboard. Similar to the 2D rotation, we train with $-45 \leq \alpha_t \leq 45$ degree rotation, we scale $\alpha_g = \alpha_t / R$ where $R = 45$, and apply a mask during training.

## A.3 LINEAR NN($z$) WALK

Rather than defining $w$ as a vector in $z$ space (Eq. 1), one could define it as a function that takes a $z$ as input and maps it to the desired $z'$ after taking a variable-sized step $\alpha$ in latent space. In this case, we may parametrize the walk with a neural network $w = \text{NN}(z)$, and transform the image using $G(z + \alpha \text{NN}(z))$. However, as we show in the following proof, this idea will not learn to let $w$ be a function of $z$.

*Proof.* For simplicity, let $w = F(z)$. We optimize for $J(w, \alpha) = \mathbb{E}_z \left[ \mathcal{L}(G(z + \alpha w), \text{edit}(G(z), \alpha)) \right]$ where $\alpha$ is an arbitrary scalar value. Note that for the target image, two equal edit operations is equivalent to performing a single edit of twice the size (e.g., shifting by 10px the same as shifting by 5px twice; zooming by 4x is the same as zooming by 2x twice). That is,

$$\text{edit}(G(z), 2\alpha) = \text{edit}(\text{edit}(G(z), \alpha), \alpha).$$

To achieve this target, starting from an initial $z$, we can take two steps of size $\alpha$ in latent space as follows:

$$z_1 = z + \alpha F(z)$$
$$z_2 = z_1 + \alpha F(z_1)$$

However, because we let $\alpha$ take on any scalar value during optimization, our objective function enforces that starting from $z$ and taking a step of size $2\alpha$ equals taking two steps of size $\alpha$:

$$z + 2\alpha F(z) = z_1 + \alpha F(z_1) \tag{7}$$

Therefore:

$$z + 2\alpha F(z) = z + \alpha F(z) + \alpha F(z_1) \Rightarrow$$
$$\alpha F(z) = \alpha F(z_1) \Rightarrow$$
$$F(z) = F(z_1).$$

Thus $F(\cdot)$ simply becomes a linear trajectory that is independent of the input $z$. $\qquad\square$

## A.4 OPTIMIZATION FOR THE NON-LINEAR WALK

Given the limitations of the previous walk, we define our nonlinear walk $F(z)$ using discrete step sizes $\epsilon$. We define $F(z)$ as $z + \text{NN}(z)$, where the neural network NN learns a fixed $\epsilon$ step transformation, rather than a variable $\alpha$ step. We then renormalize the magnitude $z$. This approach mimics the Euler method for solving ODEs with a discrete step size, where we assume that the gradient of the transformation in latent space is of the form $\epsilon \frac{dz}{dt} = \text{NN}(z)$ and we approximate $z_{i+1} = z_i + \epsilon \frac{dz}{dt}|_{z_i}$. The key difference from A.3 is the fixed step size, which avoids optimizing for the equality in (7).

We use a two-layer neural network to parametrize the walk, and optimize over 20000 samples using the Adam optimizer as before. Positive and negative transformation directions are handled with two neural networks having identical architecture but independent weights. We set $\epsilon$ to achieve the same transformation ranges as the linear trajectory within 4-5 steps.

# B  ADDITIONAL EXPERIMENTS

## B.1  MODEL AND DATA DISTRIBUTIONS

How well does the model distribution of each property match the dataset distribution? If the generated images do not form a good approximation of the dataset variability, we expect that this would also impact our ability to transform generated images. In Fig. 10 we show the attribute distributions of the BigGAN model $G(z)$ compared to samples from the ImageNet dataset. We show corresponding results for StyleGAN and its respective datasets in Appendix B.5. While there is some bias in how well model-generated images approximate the dataset distribution, we hypothesize that additional biases in our transformations come from variability in the training data.

## B.2  QUANTIFYING TRANSFORMATION LIMITS

We observe that when we increase the transformation magnitude $\alpha$ in latent space, the generated images become unrealistic and the transformation ceases to have further effect. We show this qualitatively in Fig. 3. To quantitatively verify this trends, we can compute the LPIPS perceptual distance of images generated using consecutive pairs of $\alpha_i$ and $\alpha_{i+1}$. For shift and zoom transformations, perceptual distance is larger when $\alpha$ (or $\log(\alpha)$ for zoom) is near zero, and decreases as the the magnitude of $\alpha$ increases, which indicates that large $\alpha$ magnitudes have a smaller transformation effect, and the transformed images appear more similar. On the other hand, color and rotate in 2D/3D exhibit a steady transformation rate as the magnitude of $\alpha$ increases.

Note that this analysis does not tell us how well we achieve the specific transformation, nor whether the latent trajectory deviates from natural-looking images. Rather, it tells us how much we manage to change the image, regardless of the transformation target. To quantify how well each transformation is achieved, we rely on attribute detectors such as object bounding boxes (see B.3).

## B.3  DETECTED BOUNDING BOXES

To quantify the degree to which we are able to achieve the zoom and shift transformations, we rely on a pre-trained *MobileNet-SSD v1*[3] object detection model. In Fig. 12 and 13 we show the results of applying the object detection model to images from the dataset, and images generated by the model under the zoom, horizontal shift, and vertical shift transformations for randomly selected values of $\alpha$, to qualitatively verify that the object detection boundaries are reasonable. Not all ImageNet images contain recognizable objects, so we only use ImageNet classes containing objects recognizable by the detector for this analysis.

## B.4  ALTERNATIVE WALKS IN BIGGAN

### B.4.1  LPIPS OBJECTIVE

In the main text, we learn the latent space walk $w$ by minimizing the objective function:

$$J(w, \alpha) = \mathbb{E}_z \left[ \mathcal{L}(G(z + \alpha w), \texttt{edit}(G(z), \alpha)) \right]. \tag{8}$$

using a Euclidean loss for $\mathcal{L}$. In Fig. 14 we show qualitative results using the LPIPS perceptual similarity metric (Zhang et al., 2018) instead of Euclidean loss. Walks were trained using the same parameters as those in the linear-L2 walk shown in the main text: we use 20k samples for training, with Adam optimizer and learning rate 0.001 for zoom and color, 0.0001 for the remaining edit operations (due to scaling of $\alpha$).

### B.4.2  NON-LINEAR WALKS

Following B.4.2, we modify our objective to use discrete step sizes $\epsilon$ rather than continuous steps. We learn a function $F(z)$ to perform this $\epsilon$-step transformation on given latent code $z$, where $F(z)$ is parametrized with a neural network. We show qualitative results in Fig. 15. We perform the same set of experiments shown in the main text using this nonlinear walk in Fig. 16. These experiments

---

[3]https://github.com/opencv/opencv/wiki/TensorFlow-Object-Detection-API

exhibit similar trends as we observed in the main text – we are able to modify the generated distribution of images using latent space walks, and the amount to which we can transform is related to the variability in the dataset. However, there are greater increases in FID when we apply the non-linear transformation, suggesting that these generated images deviate more from natural images and look less realistic.

### B.4.3 ADDITIONAL QUALITATIVE EXAMPLES

We show qualitative examples for randomly generated categories for BigGAN linear-L2, linear LPIPS, and nonlinear trajectories in Figs. 17, 18, 19 respectively.

### B.5 WALKS IN STYLEGAN

We perform similar experiments for linear latent space walks using StyleGAN models trained on the LSUN cat, LSUN car, and FFHQ face datasets. As suggested by Karras et al. (2018), we learn the walk vector in the intermediate $W$ latent space due to improved attribute disentanglement in $W$. We show qualitative results for color, shift, and zoom transformations in Figs. 20, 22, 24 and corresponding quantitative analyses in Figs. 21, 23, 25. We show qualitative examples for the comparison of optimizing in the $W$ and $z$ latent spaces in Stylegan in 28.

### B.6 WALKS IN PROGRESSIVE GAN

We also experiment with the linear walk objective in the latent space of Progressive GAN Karras et al. (2017). One interesting property of the Progressive GAN interpolations is that they take much longer to train to have a visual effect – for example for color, we could obtain drastic color changes in Stylegan W latent space using as few as 2k samples, but with progressive gan, we used 60k samples and still did not obtain as strong of an effect. This points to the Stylegan w latent space being more "flexible" and generalizable for transformation, compared to the latent space of progressive GAN. Moreover, we qualitatively observe some entanglement in the progressive gan transformations – for example, changing the level of zoom also changes the lighting. We did not observe big effects in the horizontal and vertical shift transformations. Qualitative examples and quantitative results are shown in Figs. 26, 27.

### B.7 QUALITATIVE EXAMPLES FOR ADDITIONAL TRANSFORMATIONS

Since the color transformation operates on individual pixels, we can optimize the walk using a segmented target – for example when learning a walk for cars, we only modify pixels in segmented car region when generating `edit`$(G(z), \alpha)$. StyleGAN is able to roughly localize the color transformation to this region, suggesting disentanglement of different objects within the $W$ latent space (Fig. 29 left) as also noted in Karras et al. (2018); Shen et al. (2019). We also show qualitative results for adjust image contrast (Fig. 29 right), and for combining zoom, shift X, and shift Y transformations (Fig. 30).

### B.8 ADDITIONAL RESULTS FOR IMPROVING MODEL STEERABILITY

We further test the hypothesis that dataset variability impacts the amount we are able to transform by comparing DCGAN models trained with and without data augmentation. Namely, with data augmentation, the discriminator is able to see edited versions of the real images. We also jointly train the model and the walk trajectory which encourages the model to learn linear walks. For zoom, horizontal shift, and 2D rotate transformations, additional samples for three training approaches – without data augmentation, with data augmentation, and joint optimization – appear in Fig. 31-33. Qualitatively, transformations using the model trained without data augmentation degrade the digit structure as $\alpha$ magnitude increases, and may even change one digit to another. Training with data augmentation and joint optimization better preserves digit structure and identity.

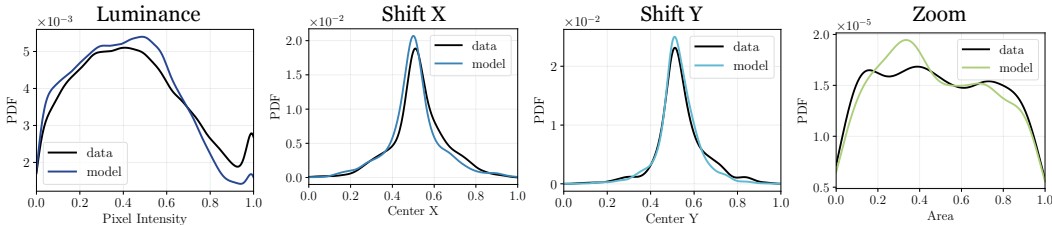

Figure 10: Comparing model versus dataset distribution. We plot statistics of the generated under the color (luminance), zoom (object bounding box size), and shift operations (bounding box center), and compare them to the statistics of images in the training dataset.

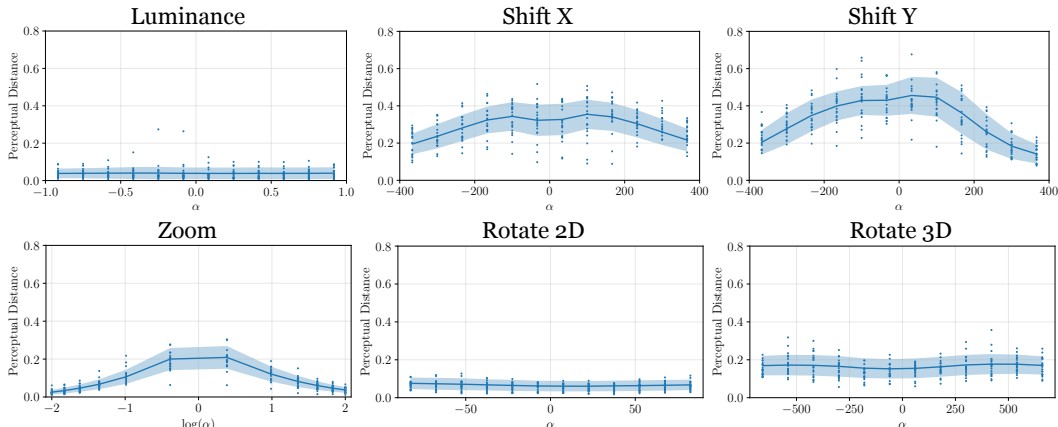

Figure 11: LPIPS Perceptual distances between images generated from pairs of consecutive $\alpha_i$ and $\alpha_{i+1}$. We sample 1000 images from randomly selected categories using BigGAN, transform them according to the learned linear trajectory for each transformation. We plot the mean perceptual distance and one standard deviation across the 1000 samples (shaded area), as well as 20 individual samples (scatterplot). Because the Rotate 3D operation undershoots the targeted transformation, we observe more visible effects when we increase the $\alpha$ magnitude.

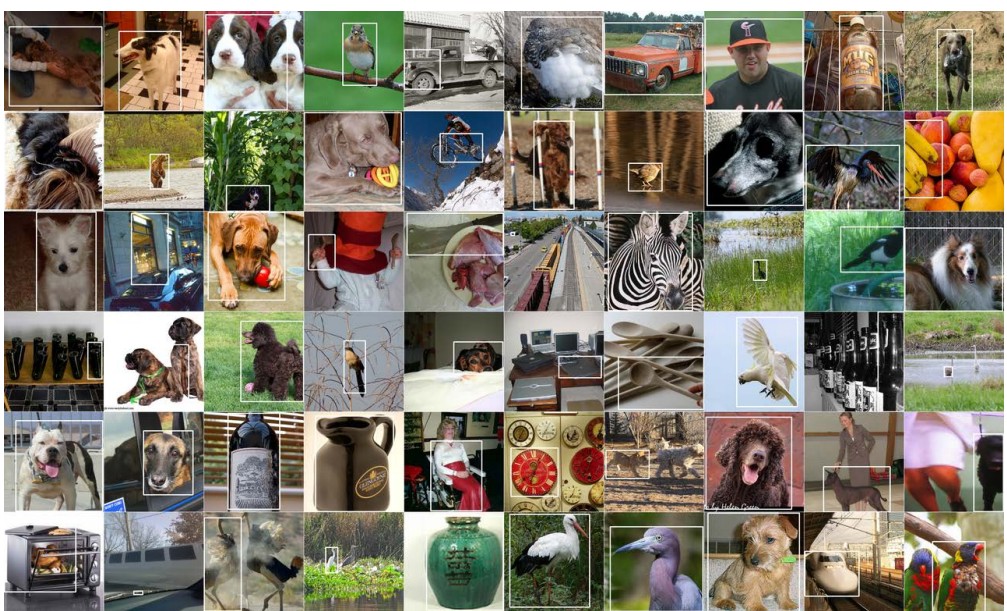

Figure 12: Bounding boxes for random selected classes using ImageNet training images.

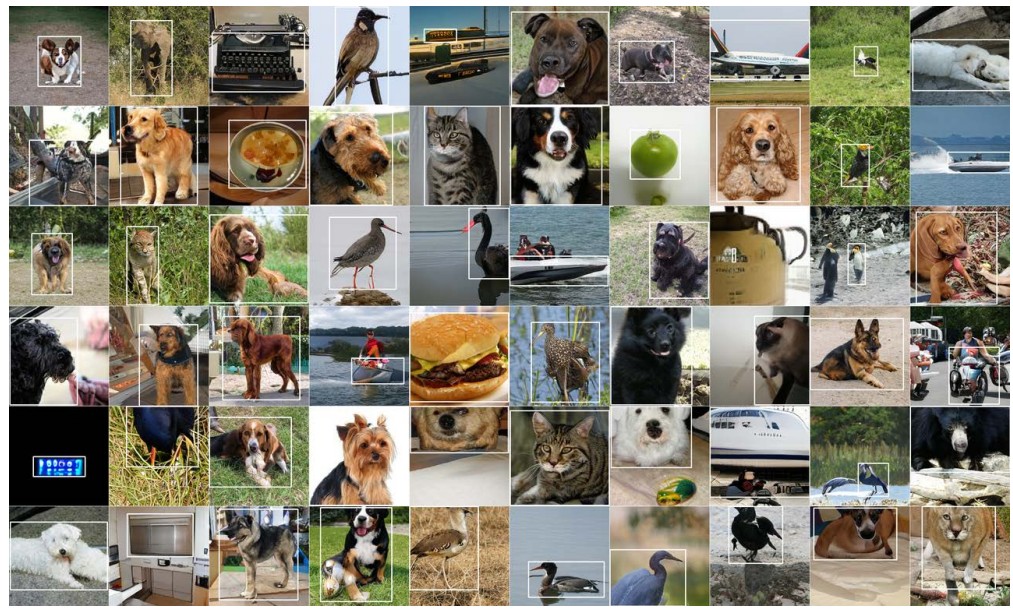

Figure 13: Bounding boxes for random selected classes using model-generated images for zoom and horizontal and vertical shift transformations under random values of $\alpha$.

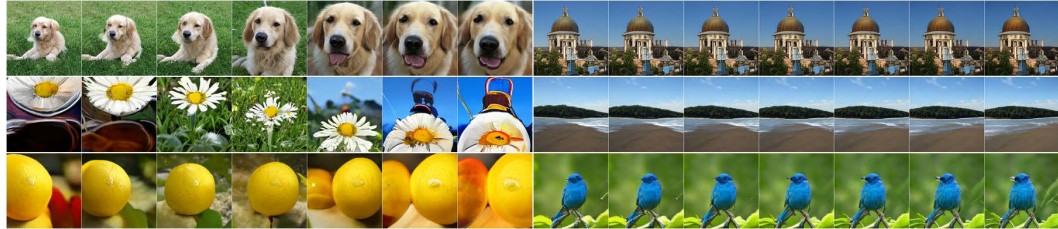

Figure 14: Linear walks in BigGAN, trained to minimize LPIPS loss. For comparison, we show the same samples as in Fig. 1 (which used a linear walk with L2 loss).

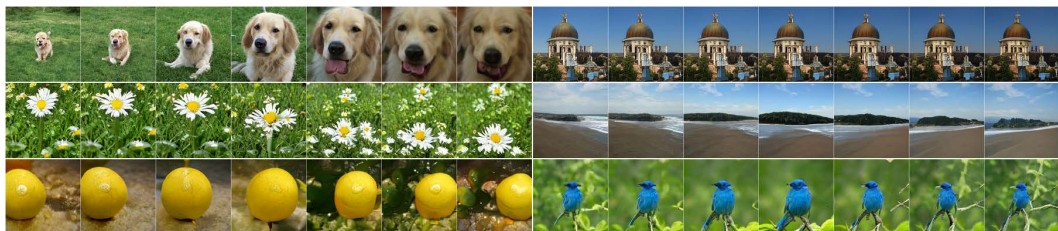

Figure 15: Nonlinear walks in BigGAN, trained to minimize L2 loss for color and LPIPS loss for the remaining transformations. For comparison, we show the same samples in Fig. 1 (which used a linear walk with L2 loss), replacing the linear walk vector $w$ with a nonlinear walk.

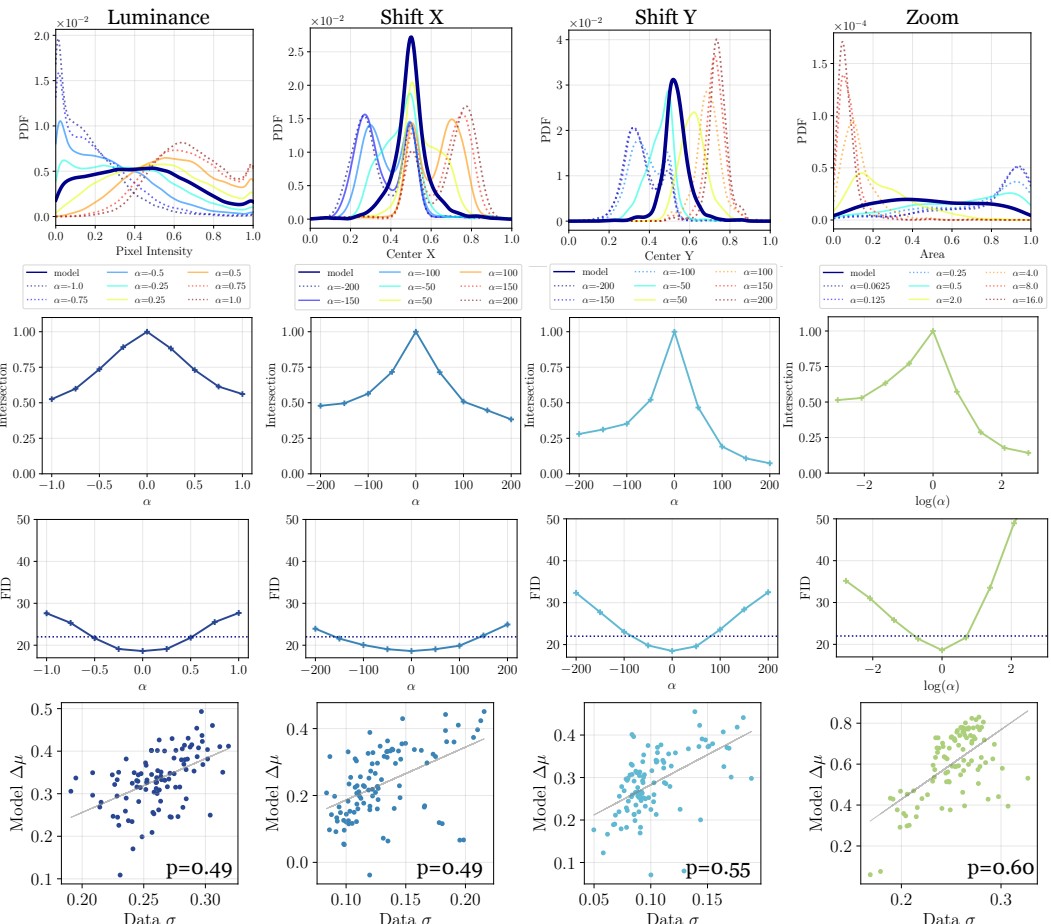

Figure 16: Quantitative experiments for nonlinear walks in BigGAN. We show the attributes of generated images under the raw model output $G(z)$, compared to the distribution under a learned transformation $\text{model}(\alpha)$, the intersection area between $G(z)$ and $\text{model}(\alpha)$, FID score on transformed images, and scatterplots relating dataset variability to the extent of model transformation.

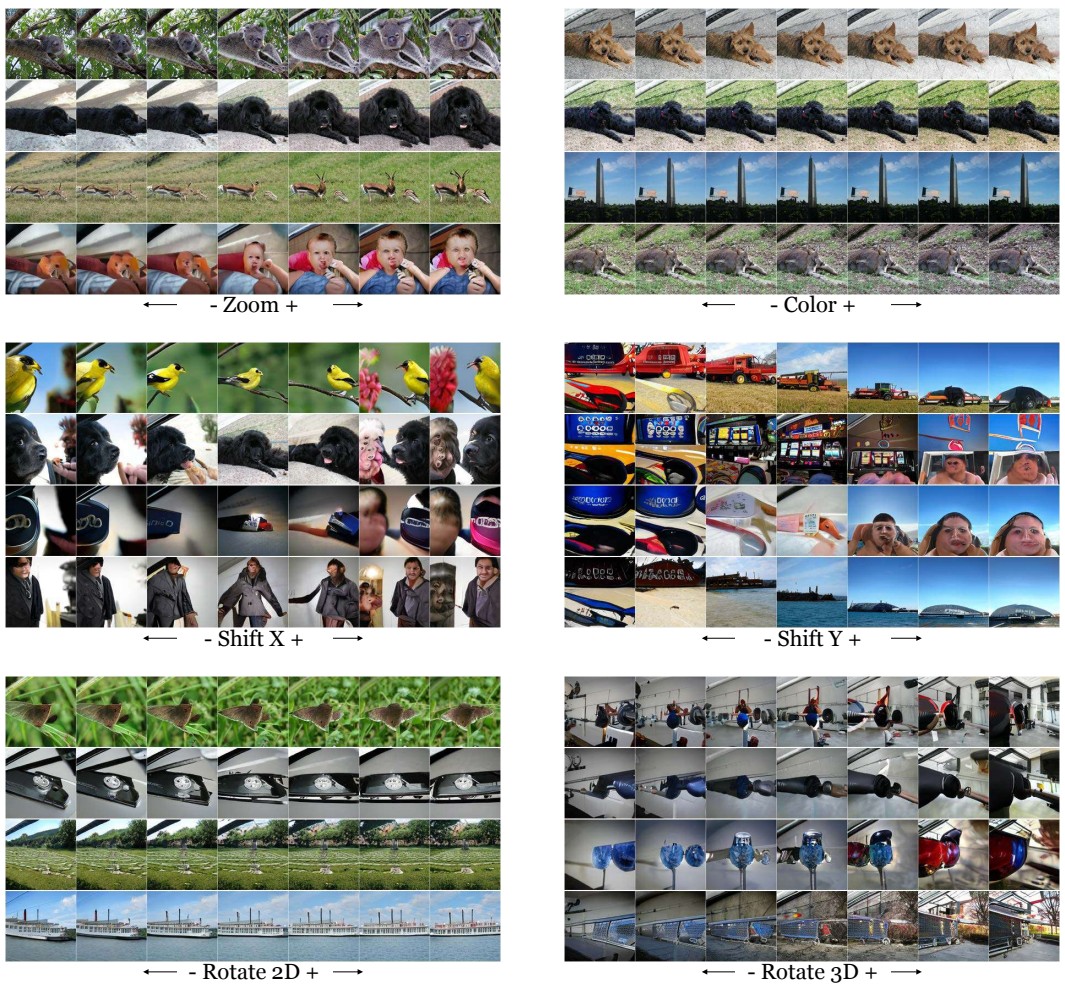

Figure 17: Qualitative examples for randomly selected categories in BigGAN, using the linear trajectory and L2 objective.

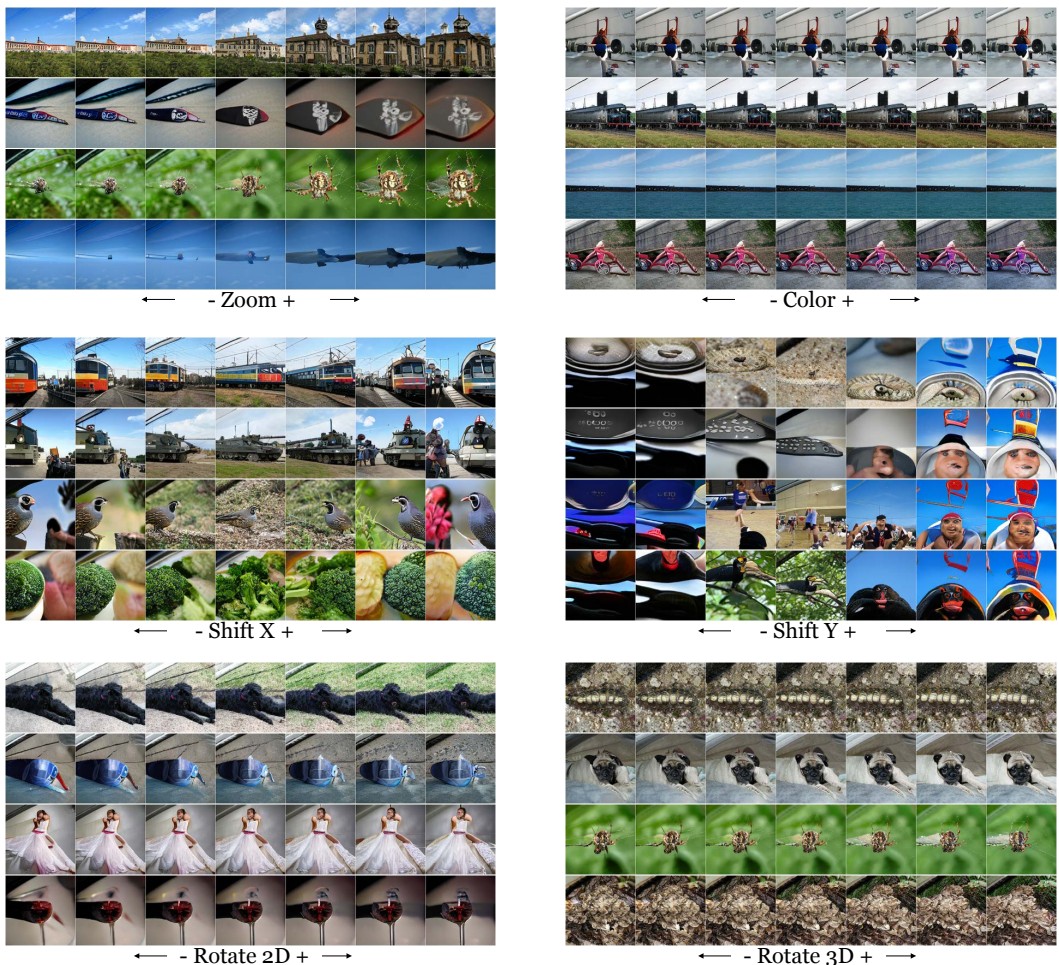

Figure 18: Qualitative examples for randomly selected categories in BigGAN, using the linear trajectory and LPIPS objective.

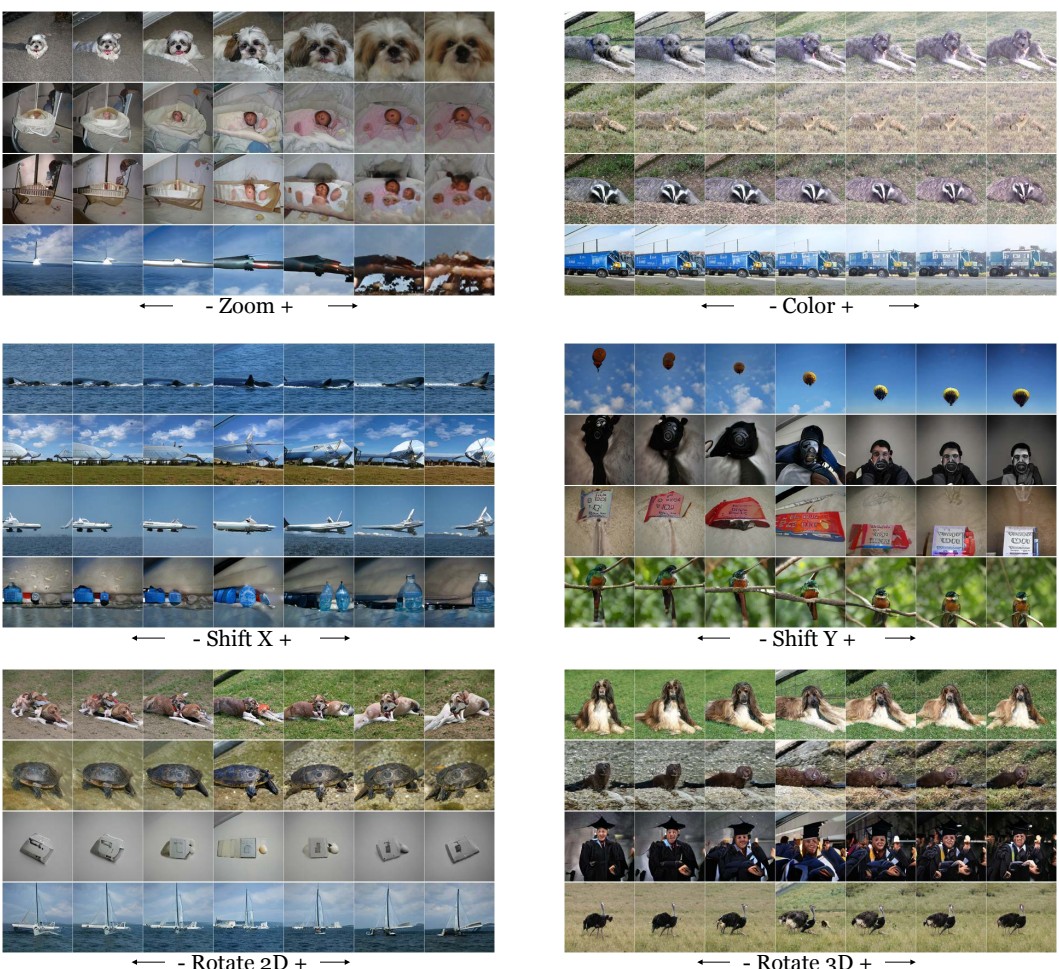

Figure 19: Qualitative examples for randomly selected categories in BigGAN, using a nonlinear trajectory.

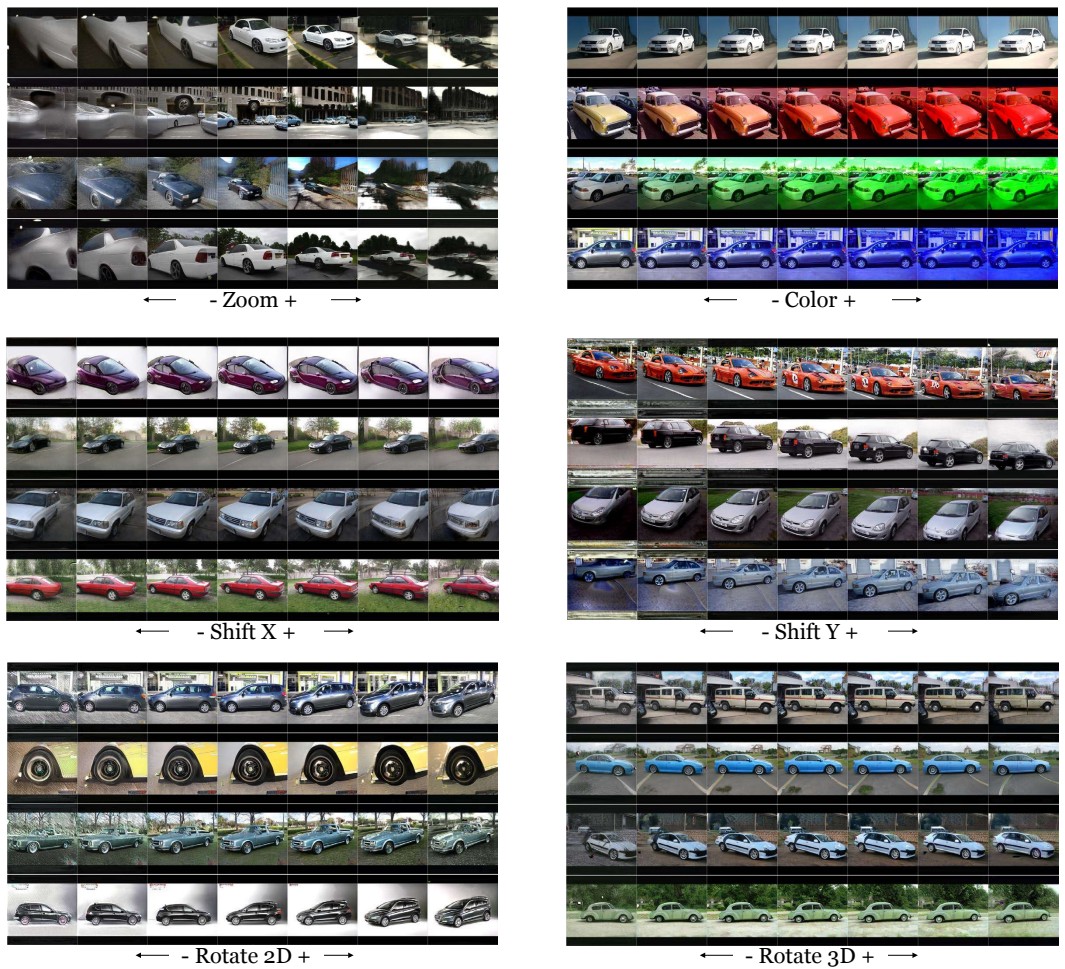

Figure 20: Qualitative examples for learned transformations using the StyleGAN car generator.

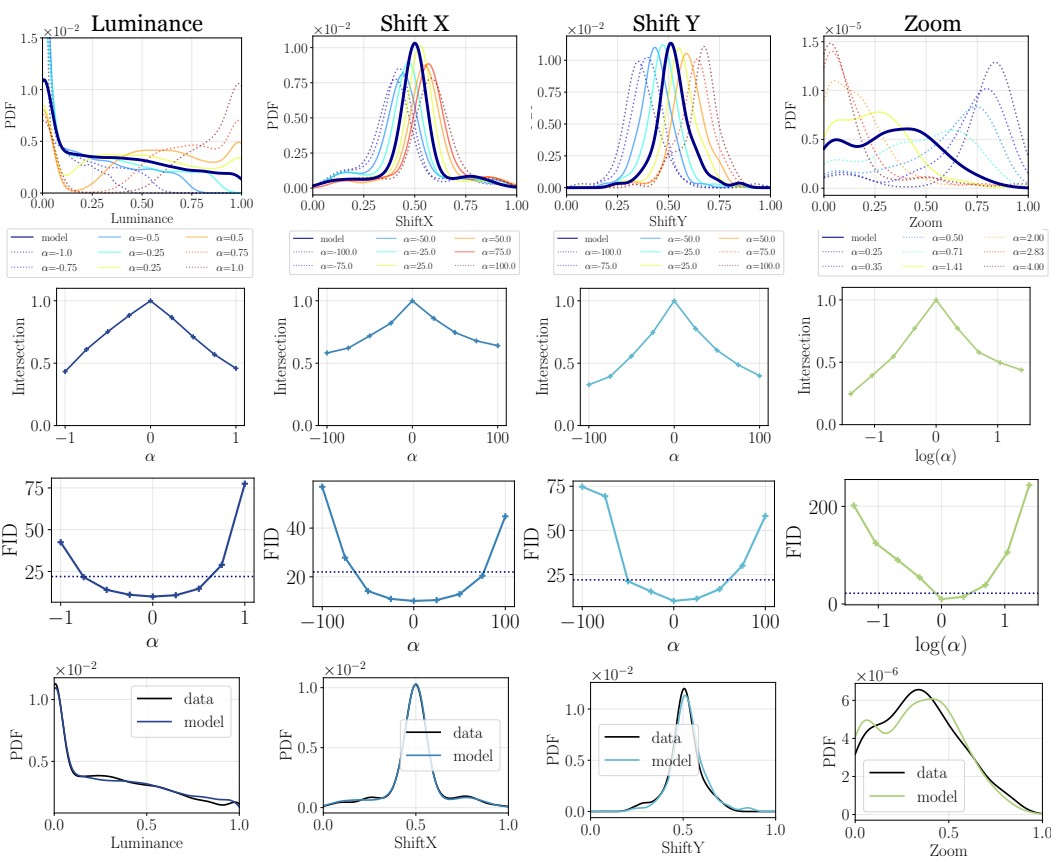

Figure 21: Quantitative experiments for learned transformations using the StyleGAN car generator.

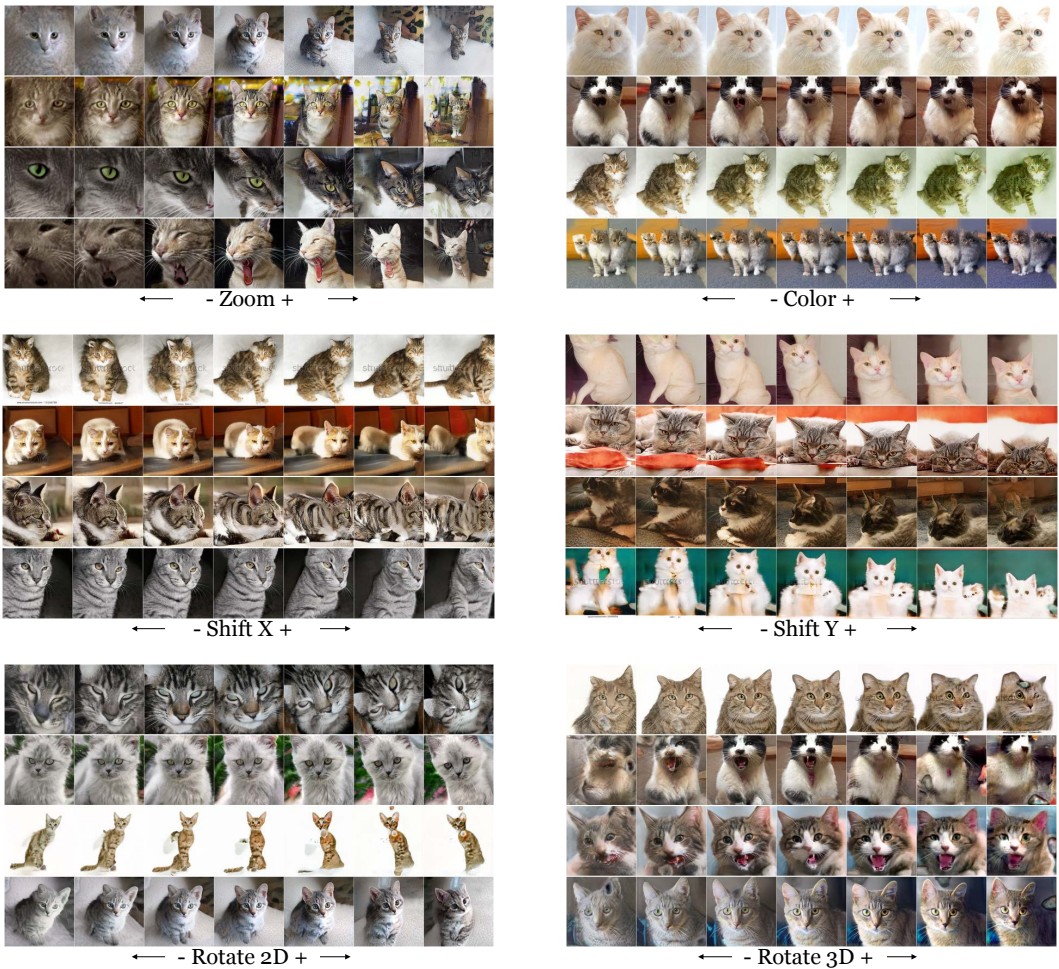

Figure 22: Qualitative examples for learned transformations using the StyleGAN cat generator.

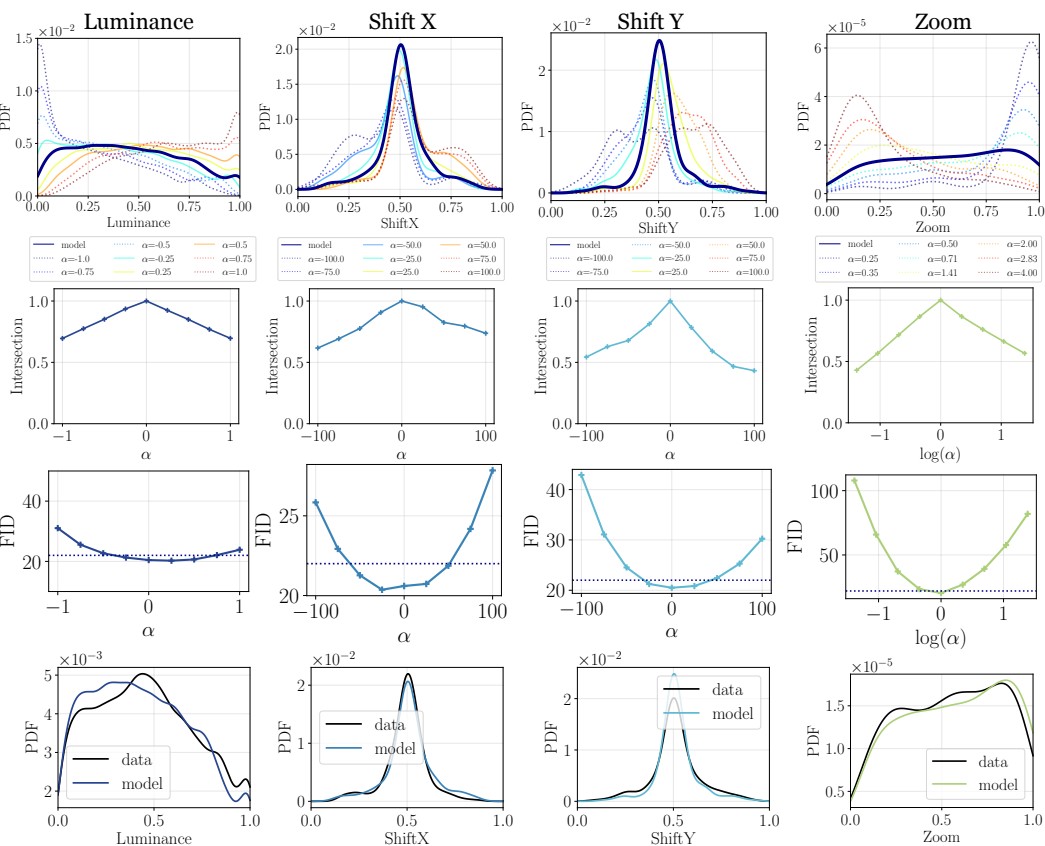

Figure 23: Quantitative experiments for learned transformations using the StyleGAN cat generator.

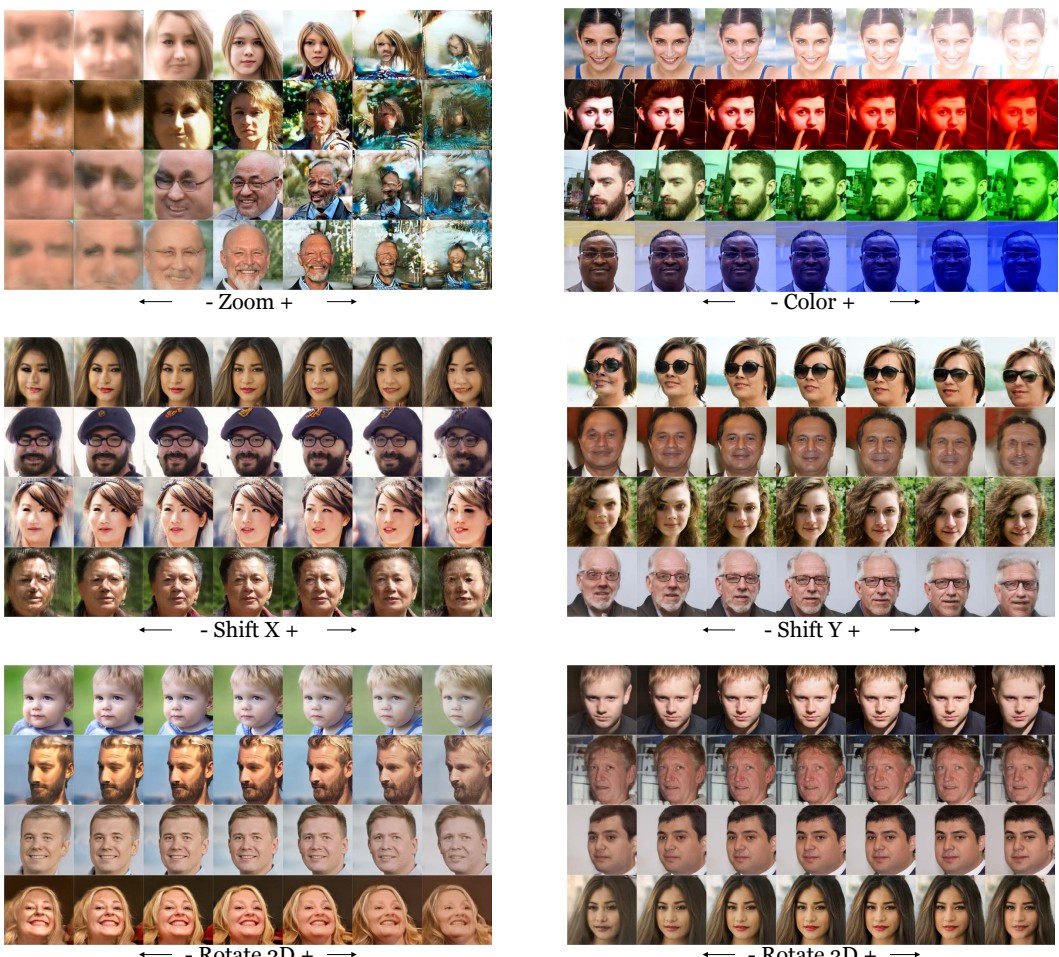

Figure 24: Qualitative examples for learned transformations using the StyleGAN FFHQ face generator.

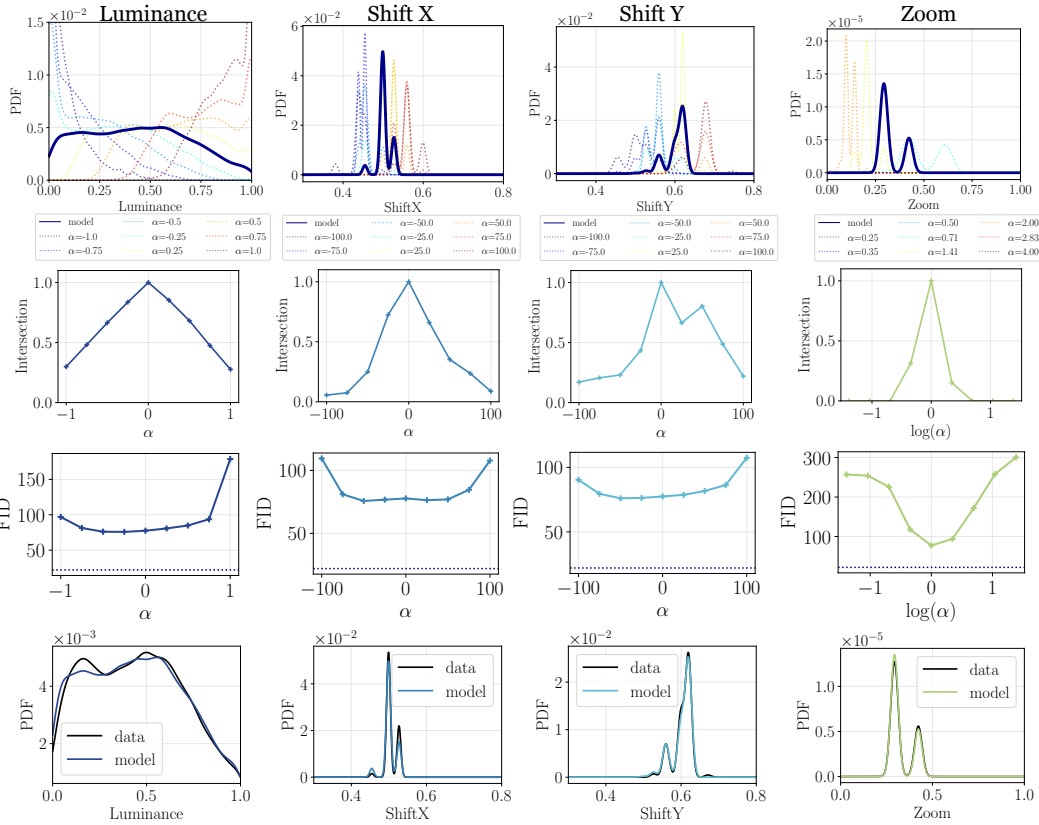

Figure 25: Quantitative experiments for learned transformations using the StyleGAN FFHQ face generator. For the zoom operation not all faces are detectable; we plot the distribution as zeros for $\alpha$ values in which no face is detected. We use the dlib face detector (King, 2009) for bounding box coordinates.

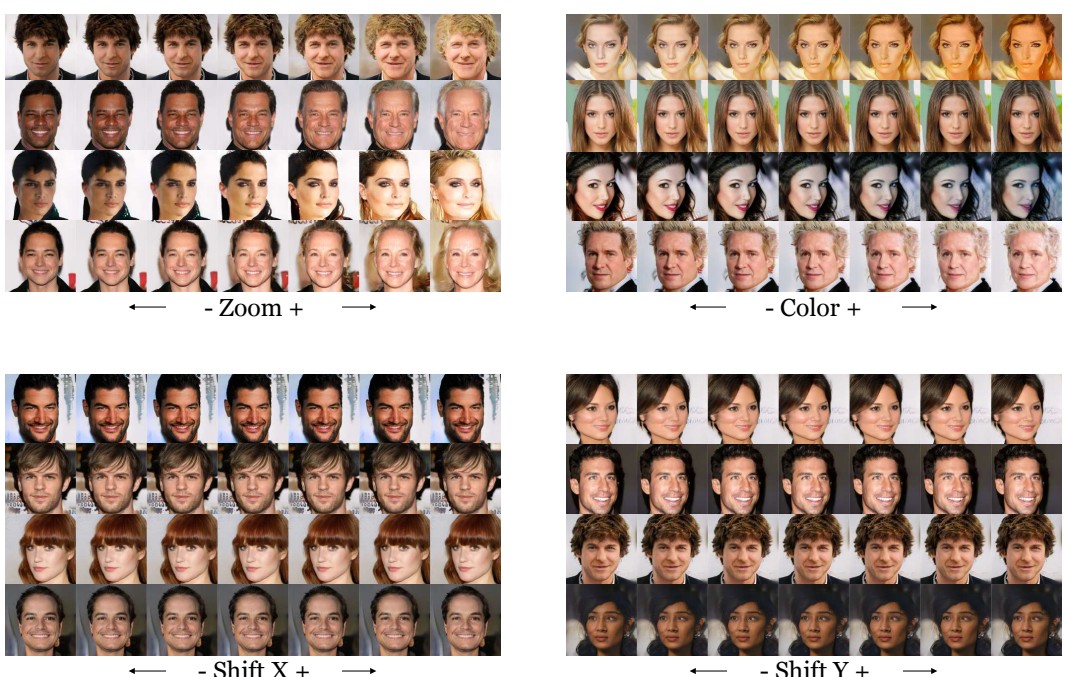

Figure 26: Qualitative examples for learned transformations using the Progressive GAN CelebaA-HQ face generator.

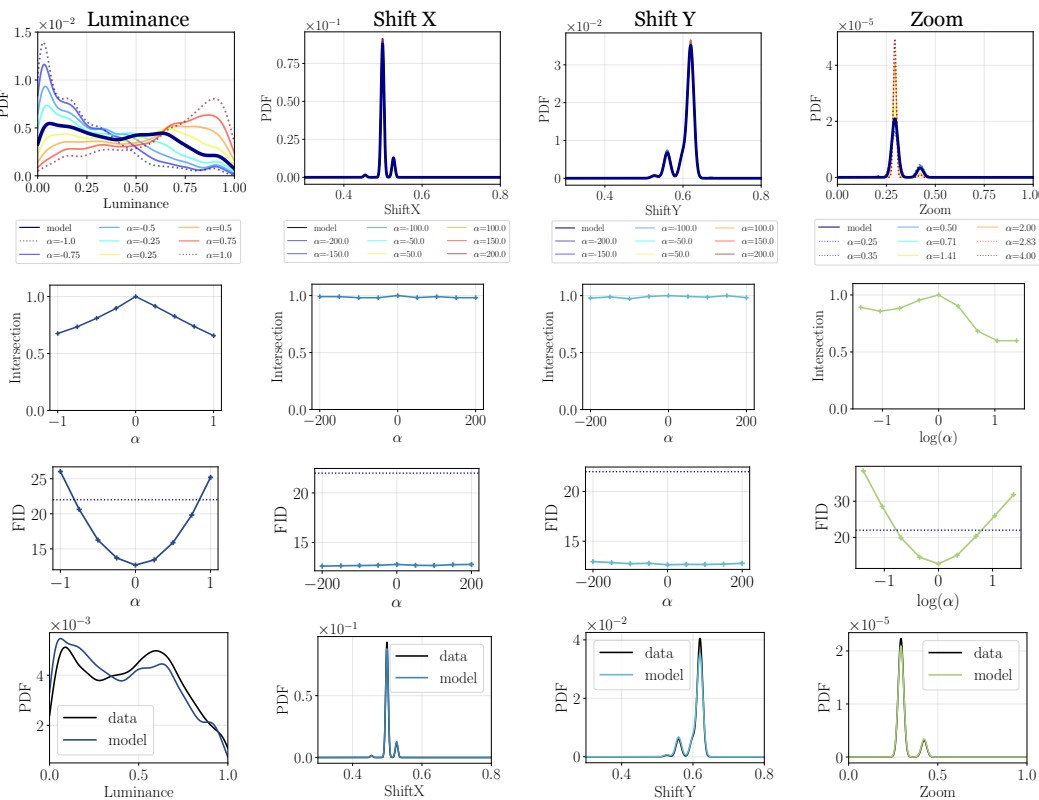

Figure 27: Quantitative experiments for learned transformations using the Progressive GAN CelebA-HQ face generator.

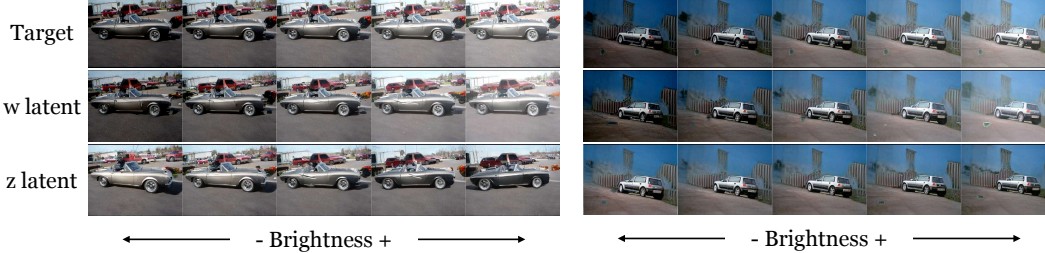

Figure 28: Comparison of optimizing for color transformations in the Stylegan w and z latent spaces.

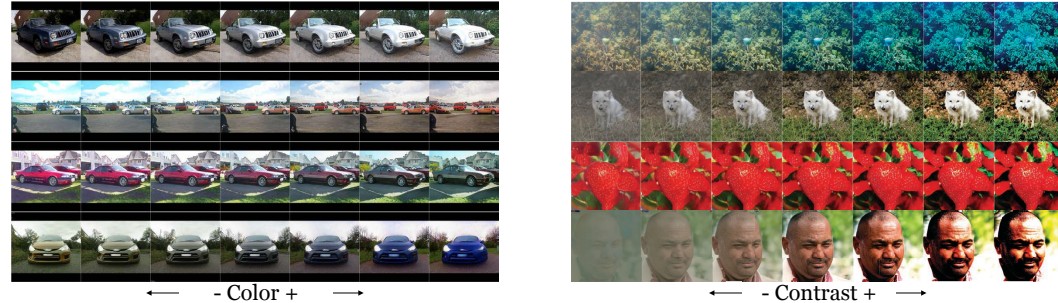

Figure 29: Qualitative examples of optimizing for a color walk with a segmented target using Style-GAN in left column and a contrast walk for both BigGAN and StyleGAN in the right column.

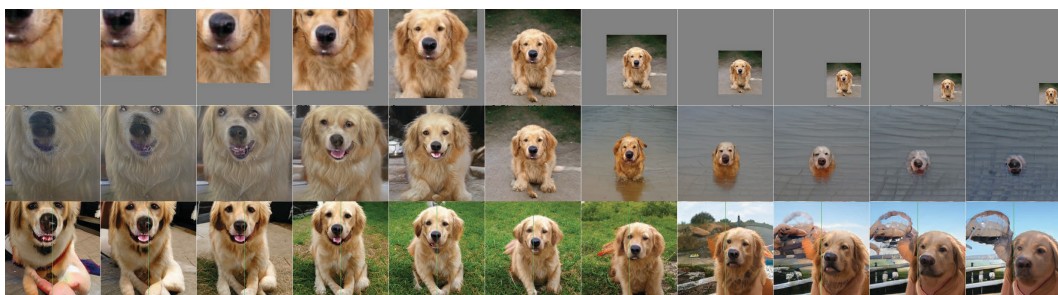

Figure 30: Qualitative examples of a linear walk combining the zoom, shift X, and shift Y transformations. First row shows the target image, second row shows the result of learning a walk for the three transformations jointly, and the third row shows results for combining the separately trained walks. Green vertical line denotes image center.

argmin W          argmin G,W          argmin W + aug

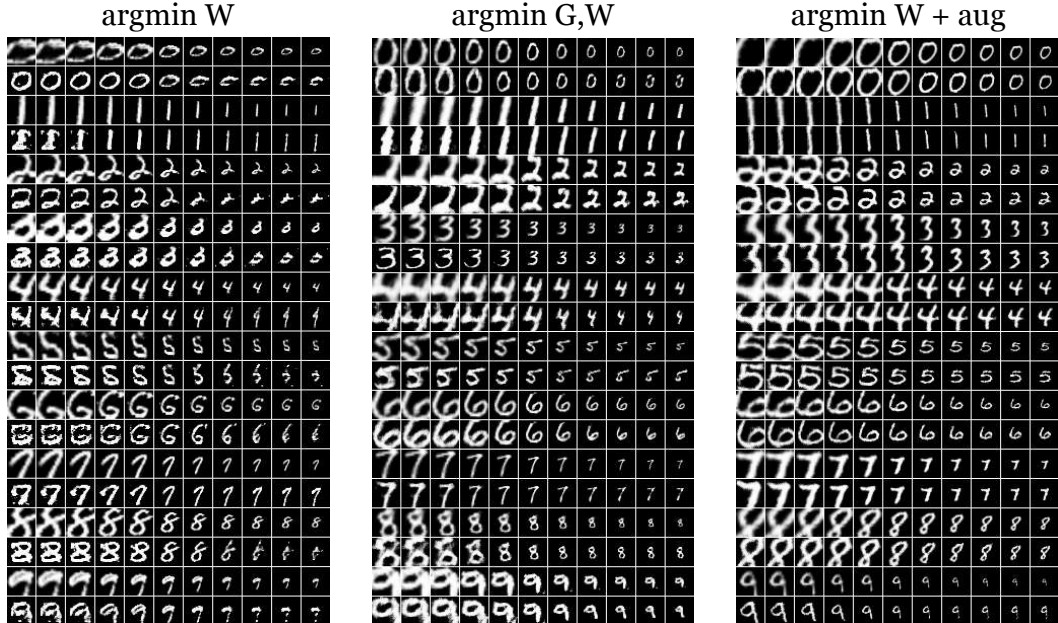

Figure 31: Quantitative experiments on steerability with an MNIST DCGAN for the Zoom transformation. Odd rows are the target images and even rows are the learned transformations.

argmin W          argmin G,W          argmin W + aug

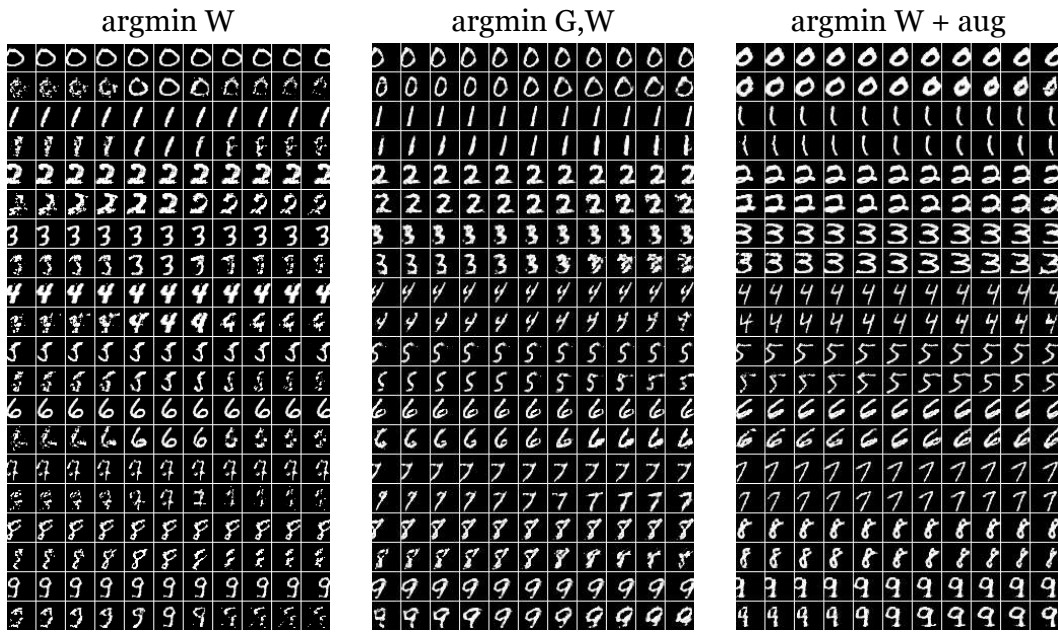

Figure 32: Quantitative experiments on steerability with an MNIST DCGAN for the Shift X transformation. Odd rows are the target images and even rows are the learned transformations.

argmin W          argmin G,W          argmin W + aug

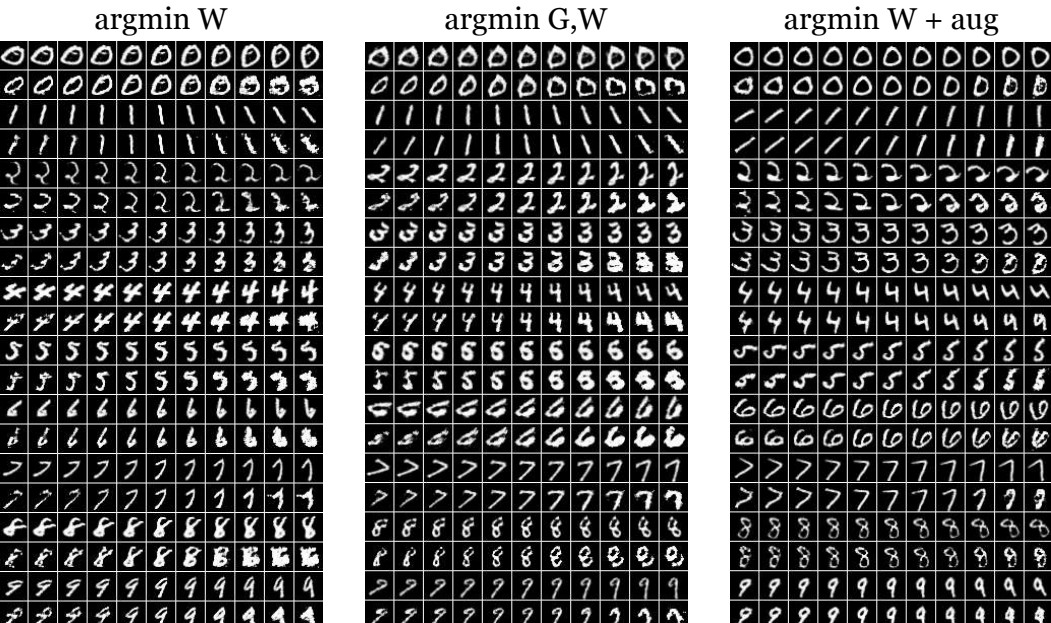

Figure 33: Quantitative experiments on steerability with an MNIST DCGAN for the Rotate 2D transformation. Odd rows are the target images and even rows are the learned transformations.

