# OpenReview forum: "On the "steerability" of generative adversarial networks"
_ICLR.cc/2020/Conference — Accept (Poster)_

### Official Review · AnonReviewer2 · 2019-10-20
**Official Blind Review #2**

**Rating:** 8

**Review:**

This work explores the extent to which the natural image manifold is captured by generative adversarial networks (GANs) by performing walks in the latent space of pretrained models. To perform these walks, a transformation vector is learned by minimizing the distance between transformed images and the corresponding images generated from transformed latent vectors. It is found that when traversing the latent space of the GAN along the direction of the transformation vector, that the corresponding generated images initially exhibit the desired transform (such as zooming or changing X position), but soon reach a limit where further changes in the latent vector do not result in changes to the image. It is observed that this behaviour is likely due to bias in the dataset which the GAN is trained on, and that by exploring the limits of the generator, biases which exist in the original dataset can be revealed. In order to increase the extents to which images can be transformed, it is shown that GANs can be trained with an augmented dataset and using a loss function that encourages transformations to lie along linear paths.

Overall, I would tend towards accepting this paper. Improving the amount of control that we have over generative models is desirable for image synthesis, and this paper does a great job of demonstrating the extent to which these models can be manipulated in terms of mimicking basic transforms. Figures are very clean and informative, and experimental results are extensive. I don't have much else to say about this paper, as I did not find anything in it that concerned me, and the paper answered all of my questions.

**Experience Assessment:**

I have published one or two papers in this area.

**Review Assessment: Checking Correctness Of Derivations And Theory:**

I did not assess the derivations or theory.

**Review Assessment: Checking Correctness Of Experiments:**

I assessed the sensibility of the experiments.

**Review Assessment: Thoroughness In Paper Reading:**

I read the paper at least twice and used my best judgement in assessing the paper.

---

> ### Author Response · Authors · 2019-11-11
> **Response to Reviewer 2**
>
> Thank you for your review and comments.

---

### Official Review · AnonReviewer1 · 2019-10-25
**Official Blind Review #1**

**Rating:** 8

**Review:**

This paper propose to study the generalization properties of GANs through interpolation. They first propose to learn a linear (and non-linear) interpolation in the latent space for a specific type of image transformation for example zoom, translation, rotation, luminance, etc... They show that linear interpolation in GANs can produce really realistic images along the path and enable to control and transform generated images to some extent. They then propose to measure to what extent the generated images can be transformed without "breaking".  Finally they show that the quality of the interpolation can be improved by learning the interpolation and generator jointly.

I'm in favour of accepting this paper. The paper is well written and organized. The experiments and observations are very interesting and really illustrate the generalization capacity of GANs.

Main argument:
- I think those observations are very valuable to the community and are a good way to get insight into the capabilities of GANs. This also give interesting informations about the different bias present and learnt in the dataset. This could also lead to very nice applications.
- The interpolation with StyleGAN and BigGAN seem to give qualitatively very different results. It would have been very interesting to study the quality of interpolations on more models and datasets, and compare their generalization capabilities as well as the bias present in the different datasets.
- Does training the generator and interpolation jointly improve the quality of the generator in general ? It would have been nice to run this method on more complicated dataset like CIFAR10 and see if this method increase the overall FID score.


Minor comments:
- In appendix A.2 the authors explain how the range of $\alpha$ is set for the different experiments. However it's not clear how is this range used in practice ? Do you sample uniformly $\alpha$ in this range to train the linear interpolation ? Also how many steps are required to learn the linear interpolation ? How much the does it influence the quality of the interpolation ?
- There is a typo in equation 6
- In figure 6: What does the right figure represent ? especially what are the different colours ?

**Experience Assessment:**

I have published in this field for several years.

**Review Assessment: Checking Correctness Of Derivations And Theory:**

N/A

**Review Assessment: Checking Correctness Of Experiments:**

I assessed the sensibility of the experiments.

**Review Assessment: Thoroughness In Paper Reading:**

I read the paper at least twice and used my best judgement in assessing the paper.

---

> ### Author Response · Authors · 2019-11-15
> **Response to Reviewer 1**
>
> Thank you for your comments and questions; we have incorporated these in the revision and respond to your questions below.
>
> Main Argument:
>
> Q1: It would have been very interesting to study the quality of interpolations on more models and datasets, and compare their generalization capabilities as well as the bias present in the different datasets.
>
> A1: We performed some additional experiments using the progressive gan generator [1] on CelebA-HQ dataset. One interesting property of the progressive gan interpolations is that they take much longer to train to have a visual effect -- for example for color, we could obtain drastic color changes in Stylegan W latent space using as few as 2k samples, but with progressive gan, we used 60k samples and still did not obtain as strong of an effect. This points to the Stylegan w latent space being more “flexible” and generalizable for transformation, compared to the latent space of progressive GAN. Moreover, we qualitatively observe some entanglement in the progressive gan transformations -- for example, changing the level of zoom also changes the lighting. We did not observe large effects for the shift transformations, although perhaps more hyperparameter tuning may improve these results. We have added a section B.6 in the appendix and figures illustrating these results.
>
> Q2: Does training the generator and interpolation jointly improve the quality of the generator in general?
>
> A2: This is an interesting question. We are in the process of investigating this hypothesis to see if sampling from both the latent space and transformation alpha can help improve sample diversity and potentially FID.  We did not yet observe an improvement in preliminary experiments on Cifar10, but the experiments are ongoing and we will add complete results on this question to the final version of the paper.
>
> Minor Comments:
> Q1: In appendix A.2 the authors explain how the range of is set for the different experiments. However it's not clear how is this range used in practice ? Do you sample uniformly in this range to train the linear interpolation ? Also how many steps are required to learn the linear interpolation ? How much the does it influence the quality of the interpolation ?
>
> A1: We pick the ranges using two criteria: qualitatively acceptable and quantitatively under a fixed threshold for FID score. We pick alpha steps uniformly within the ranges (shifts and rotations are integer steps). For training, we have between 20k and 40k samples for all models, and beyond these numbers we don’t see much improvement.
>
> Q2: There is a typo in equation 6
>
> A2: Thank you for your careful review in catching these mistakes. We have updated the typo in the revision.
>
> Q3: In figure 6: What does the right figure represent ? especially what are the different colours ?
>
> A3: The right side of the figure has three rows: the top row shows the plot of per-class zoom variability; there are two black datapoints we chose as examples to show instances of low and high variability classes. The middle row shows the distribution of the low variability datapoint (“robin” class), and the bottom row shows the distribution of the high variability datapoint (“laptop” class). On the left of these plots we show qualitative results. In each of these two plots, we show dataset, -\alpha^*, and +\alpha^* distributions with black, green, red, respectively. We have updated the revision to clarify these in the figure caption.
>
> [1] Karras, Tero, et al. "Progressive growing of gans for improved quality, stability, and variation." ICLR (2018).

---

### Official Review · AnonReviewer4 · 2019-11-05
**Official Blind Review #3**

**Rating:** 8

**Review:**

The paper explores and experiments on extrapolating attributes of images produced by GANs by manipulating their representations in latent space. Attribute manipulation is done by predicting latent space walks (linear or non-linear) and is learned in a self-supervised way by using augmented outputs of a pretrained GAN as target images.
Authors experimentally show dependence of range of possible attribute manipulations on the diversity of the dataset in terms of that attribute as well as propose techniques to improve it.

Suggested concepts are explained in a clear way with extensive experiments confirming the findings. Techniques proposed for improving "steerability" of GANs are backed up by both qualitative and quantitative analysis, although missing experiments on more sophisticated datasets than MNIST.
Overall,  I recommend to accept this paper.

Several questions I would like the authors to address to make some details more clear and the paper more complete:
1. Why the color distribution of generated images is evaluated on a sampled subset of pixels, not full images? ("Quantifying steerability" section.)
2. On Figure 6, which classes are outlying on transformation limitation / data variability plots (bottom-right corner) and how it may be explained?
3. While StyleGAN can not preserve geometry of objects for shift in location-based attributes, when walks are learned in the W space, have you experimented on manipulating those attributes with z space? What are the results?

Other minor flaws include
1. Pictures in Fig. 2 are mixed up between G(z) and G(z + \alpha w)
2. In Fig. 2 edit(G(z, \alpha)) -> edit(G(z), \alpha))
3. In eq. (2) f^n(z) -> G(f^n(z))
4. In eq. (6) +\alpha^* -> -\alpha^*

**Experience Assessment:**

I have published one or two papers in this area.

**Review Assessment: Checking Correctness Of Derivations And Theory:**

I did not assess the derivations or theory.

**Review Assessment: Checking Correctness Of Experiments:**

I assessed the sensibility of the experiments.

**Review Assessment: Thoroughness In Paper Reading:**

I read the paper at least twice and used my best judgement in assessing the paper.

---

> ### Author Response · Authors · 2019-11-11
> **Response to Reviewer 3**
>
> Thank you for your comments and questions; we have incorporated these in the revision and respond to your questions below.
>
> Q1: Why is the color distribution generated using a subset of pixels?
>
> A1: We use a random subset of pixels simply for computational feasibility. Each distribution is compiled over 1000  images, and drawing a distribution over all pixels per image increases that number by 256^2, causing the computation to be very slow. In contrast for the remaining operations, we measure one statistic per image based on the bounding box, which is fast.
>
> Q2: What are the classes in the bottom right of the transformation limitation / data variability plots?
>
> A2: The classes in the bottom right corner of the plots are wooden spoon (shift x), cleaver (shift y), and computer keyboard (zoom). These classes are more difficult for BigGAN to model accurately, and they deform easily or become unrecognizable under alpha transformations, which may prevent the object detector from reliably detecting them.
>
> Q3: What are the results of manipulations in Stylegan z latent space?
>
> A3: We experimented with manipulations in the Stylegan z space — in general the effects of these transformations are weaker, and may entangle other transformations along with the target transformation. For example, when recoloring a car using a walk in z, it may inadvertently also rotate or zoom the car. On the other hand, in the w latent space we are better able to change the desired attribute without other side effects. We have added a new qualitative figure (Fig. 28) in the appendix illustrating these differences.
>
> Minor Flaws: Thank you for your careful review in catching these mistakes. We have updated the typos in the revision. In Fig 2 we optimize for a z which approximates a shifted version of the original image x. Hence, the G(z+\alpha w) image does not exactly match the original image G(z) or the shifted edit(G(z), \alpha), but is intended to approximate the shifted image.

---

### Author Response · Authors · 2019-11-15
**Revision**

We thank all the reviewers for their comments and feedback. We made the following changes to address the reviewers’ comments:

Review #1: Added section B.6, Fig 26, 27 in the appendix for transformations in Progressive GAN.

Review #3: Added an additional Fig 28 in the appendix comparing Stylegan latent space transformations.

Minor: Corrected typos in equation 2 and 6, and Fig. 2. Added a clarifying sentence to the caption of Fig. 6.

---

### Decision · Program_Chairs · 2019-12-19

**Decision:**

Accept (Poster)

**Comment:**

All three reviewers agree that the paper provide an interesting study on the ability of generative adversarial networks to model geometric transformations and a simple practical approach to how such ability can be improved. Acceptance as a poster is recommended.